# ONLINE FRACTIONAL KNAPSACK WITH PREDICTIONS

## ABSTRACT

The well-known classical version of the online knapsack problem decides which of the arriving items of different weights and values to accept into a capacity-limited knapsack. In this paper, we consider the online fractional knapsack problem where items can be fractionally accepted. We present the first online algorithms for this problem which incorporate prediction about the input in several forms, including predictions of the smallest value chosen in the optimal offline solution, and interval predictions which give upper and lower bounds on this smallest value. We present algorithms for both of these prediction models, prove their competitive ratios, and give a matching worst-case lower bound. Furthermore, we present a learning-augmented meta-algorithm that combines our prediction techniques with a robust baseline algorithm to simultaneously achieve consistency and robustness. Finally, we conduct numerical experiments that show that our prediction algorithms significantly outperform a simple greedy prediction algorithm for the problem and the robust baseline algorithm, which does not use predictions. Furthermore, we show that our learning-augmented algorithms can leverage imperfect predictions (e.g., from a machine learning model) to greatly improve average-case performance without sacrificing worst-case guarantees.

## 1 INTRODUCTION

In the classic online knapsack problem (OKP), the goal is to pack a finite number of sequentially arriving items with different values and weights into a knapsack with a limited capacity such that the total value of admitted items is maximized. In the online setting, the decision-maker should immediately and irrevocably admit or reject an item upon its arrival without knowing the value and weight of the future items. From a practical perspective, OKP captures a broad range of dynamic pricing and resource allocation problems in application domains such as online advertising (Zhou et al., 2008), cloud pricing and job scheduling (Zhang et al., 2017), and admission control and routing in communication networks (Buchbinder & Naor, 2009).

The basic OKP and its variants have been studied in the context of competitive analysis. Under the worst-case analysis, online algorithms are designed to minimize the *competitive ratio* (Borodin et al., 1992), which is the worst-case ratio of the profits obtained from the offline optimum and the online algorithm. It is well known that no deterministic online algorithms can achieve bounded competitive ratios (Marchetti-Spaccamela & Vercellis, 1995) for the general *integral* OKP, where items are either fully accepted or rejected. Due to this negative result, existing works mainly approach OKP in three ways: (i) considering a sub-class of knapsack problems, e.g., unit density items (Ma et al., 2019), small weight items (Zhou et al., 2008); (ii) switching to relaxed problem settings, e.g., removable items (Böckenhauer et al., 2020), resource augmentation (Böckenhauer et al., 2014b); and (iii) making additional assumptions on the input, e.g., assuming bounded value-to-weight ratios (Zhou et al., 2008; Sun et al., 2021a; Yang et al., 2021).

In this paper, we focus on an important sub-class of OKP, the online fractional knapsack problem (OFKP), where an algorithm can accept any fraction of an item. In practice, OFKP models significant real-world applications in dynamic resource allocation scenarios where each item (i.e., request) can be arbitrarily divided. Noted applications of OFKP include distributing online sequential tasks with different computational requirements to available resources efficiently Liu et al. (2011), routing varied-rate traffic through a network concerning capacity constraints Cao et al. (2022), and scheduling energy in smart grids Sun et al. (2021b).

From a theoretical perspective, fractional knapsack is well understood in the offline setting (Ishii et al., 1977; Ferdosian et al., 2015), (where a greedy algorithm solves the problem optimally), but it remains relatively understudied in the online setting. The most recent result by (Sun et al., 2021b) shows that if we additionally assume that the unit values of all items are bounded within $[L, U]$, a threshold-based algorithm for OFKP can achieve the optimal competitive ratio of $O(\ln(^U/_L))$ (see Section 2.2 for more detail). However, the bound on the unit values often gives rather coarse uncertainty quantification for the instances of OFKP and leads to poor performance when the instance is not adversarial.

Introduced by (Lykouris & Vassilvtiskii, 2018; Purohit et al., 2018), *learning-augmented algorithm design* is a framework which has gained traction as a method to leverage machine-learned predictions in algorithms without sacrificing worst-case competitive guarantees. Under this framework, online algorithms are evaluated using the concepts of *consistency* and *robustness*, which give the competitive performance when the advice is accurate or completely wrong, respectively. Recent work (Im et al., 2021; Zeynali et al., 2021; Lechowicz et al., 2023b; Böckenhauer et al., 2014b) has explored OKP with advice or machine-learned predictions (see Appendix A.1 for a comprehensive review). However, to the best of our knowledge, none of the existing works consider OFKP with prediction. The most relevant work Sun et al. (2021a) considers a special case of OFKP (referred to as one-way trading) with predictions, in which the weights of items are all equal to the knapsack capacity. In this case, a prediction of the maximum unit value is sufficient to guide the design of the learning-augmented algorithm since the offline optimal only accepts the item with the maximum value. However, in OFKP, the admission of each item is upper bounded by the item's weight (revealed online); thus, the prediction model in Sun et al. (2021a) cannot be generalized to OFKP.

Acknowledging the existing research gap in the literature, this paper focuses on the design and analysis of competitive algorithms for OFKP with advice. Our contributions are twofold. First, we introduce a new prediction model (in the form of a simple prediction about a "critical unit value") for OFKP, and design an optimal algorithm with predictions that is shown to achieve a matching lower bound (see Theorem 3.1 and Theorem 3.4). Further, we generalize this algorithm by considering two imperfect prediction models for OFKP. The first such model considers predictions of an interval (as opposed to a single critical value), and the second such model considers predictions which are probabilistically correct. We show learning-augmented algorithms for both of these models that use imperfect predictions to improve performance in the average-case while maintaining worst-case guarantees (See Theorem 4.1 and Lemma 4.3). Besides theoretical analysis, we also evaluate the empirical performance of our algorithms in numerical experiments compared against baseline algorithms without prediction, showing that our algorithms significantly outperform baseline results for OFKP and can gracefully handle errors in the prediction.

Further, we develop novel technical approaches in both algorithm design and analysis to achieve the above theoretical results. Our proposed algorithm achieves constant competitive ratios in contrast to the classic result $O(\ln(^U/_L))$, which depends on the upper and lower bounds of the unit value. To achieve this, we strategically utilize thresholds to limit the selection of high-value items and reserve capacity for units with critical values, mimicking the choices made by an optimal algorithm. Moreover, our approach goes beyond worst-case scenarios, attaining a competitive ratio approaching 1. We achieve this by employing a "prebuying" strategy. Initially, we prioritize high-value items, and subsequently adjust the knapsack's capacity to accommodate lower-value items. This allows us to optimize selections by adapting our capacity allocation in favor of high-value items during the initial stages.

## 2 PROBLEM FORMULATION, PRELIMINARIES, AND PREDICTION MODEL

### 2.1 ONLINE FRACTIONAL KNAPSACK PROBLEM

In the online fractional knapsack problem (OFKP), there is a knapsack with a capacity of 1 (WLOG, since otherwise, all weights can be scaled down). Items arrive online, each with two properties: unit value $(v_i)$ and maximum weight $(w_i)$. In the $i^{th}$ step, an online algorithm must select some portion $x_i \leq w_i$ of the $i^{th}$ item to add to the knapsack. This decision must be based only on all items seen so far, $(v_1, w_1), \ldots, (v_i, w_i)$, and is irrevocable. The algorithm obtains profit $x_i v_i$ if it admits an $x_i$ portion of the item into the knapsack. The objective is to maximize the total profit subject to the

---

**Algorithm 1** TA: An online threshold-based algorithm for OFKP without prediction

---

1: **input**: threshold function $\phi(z)$;
2: **output**: online decision $x_i$;
3: **initialization**: knapsack utilization $z^{(0)} = 0$;
4: **while** item $i$ (with unit value $v_i$ and weight $w_i$) arrives **do**
5:     **if** $v_i < \phi(z^{(i-1)})$ **then**
6:         $x_i = 0$;
7:     **else if** $v_i \geq \phi(z^{(i-1)})$ **then**
8:         $x_i = \min\{\phi^{-1}(v_i) - z^{(i-1)}, w_i, 1 - z^{(i-1)}\}$;
9:     update $z^{(i)} = z^{(i-1)} + x_i$.

---

knapsack's capacity. The offline version problem can be formulated as the following linear program:

$$\max \sum_{i=1}^{n} x_i \cdot v_i, \quad \text{s.t.} \quad \sum_{i=1}^{n} x_i \leq 1 \quad \text{and} \quad 0 \leq x_i \leq w_i \leq 1 \quad \forall i \in [n]. \tag{1}$$

We let $U$ and $L$ denote the maximum and minimum unit values for an instance. Note that these bounds are *unknown* to our algorithms and only used for analysis. This is in contrast to most existing works that assume $U$ and $L$ are known in advance (Zhou et al., 2008; Sun et al., 2021b).

## 2.2 PRIOR RESULTS: COMPETITIVE ALGORITHMS WITHOUT PREDICTION

*Competitive ratio.* OFKP has received considerable attention within the framework of competitive analysis. The primary goal is to design an online algorithm that, on every possible input instance, achieves a profit that is a large fraction of the optimum (Borodin et al., 1992). We denote $\text{OPT}(\mathcal{I})$ as the offline optimum on the input $\mathcal{I}$, and $\text{ALG}(\mathcal{I})$ represents the profit obtained by an online algorithm (ALG) on that input. If ALG is randomized, then we define $\text{ALG}(\mathcal{I})$ to be the expected profit on input instance $\mathcal{I}$. Formally, let $\Omega$ denote the set of all possible inputs, the competitive ratio (CR) of an online algorithm is defined as $\text{CR} = \max_{\mathcal{I} \in \Omega} \text{OPT}(\mathcal{I})/\text{ALG}(\mathcal{I})$. Observe that CR is greater than or equal to one. The smaller it is, the more effectively the algorithm performs.

*State-of-the-art results.* OFKP has seen relatively little attention in the literature despite being a classic relaxation of the integral knapsack problem. Most results for OFKP make different assumptions such as random ordering of the input (Giliberti & Karrenbauer, 2021) or introduce additional components such as removable items (Noga & Sarbua, 2005). A recent result by Sun et al. (2021b) is the closest to our setting, showing that if the unit value is bounded, i.e., $v_i \in [L, U], \forall i \in [n]$, a threshold-based algorithm can achieve the optimal competitive ratio among all online algorithms.

The threshold-based algorithm is shown in Algorithm 1. This algorithm takes a threshold function $\phi(z) : [0, 1] \rightarrow [L, U]$ as its input. Specifically, $\phi(z)$ can be understood as the pseudo price of packing a small amount of item when the knapsack's current *utilization* (i.e. the fraction of knapsack's total capacity which is filled with previously accepted items) is $z$. The algorithm rejects an item $i$ if its unit value $v_i$ is smaller than the pseudo price $\phi(z^{(i-1)})$ at the current utilization $z^{(i-1)}$. Otherwise, the algorithm will continuously admit the item until one of the following three cases occurs: (i) the utilization reaches $\phi^{-1}(v_i)$ (i.e., the pseudo price reaches $v_i$); (ii) the entire item is admitted; or (iii) the knapsack capacity is used up. Notice the threshold function $\phi$ is the only design space for Algorithm 1. Sun et al. (2021b) shows that the optimal competitive ratio can be attained when $\phi$ is carefully designed as follows.

**Lemma 2.1** (Theorem 3.5 & 3.6 in Sun et al. (2021b)). *For* OFKP*, if the unit value of items is bounded within* $[L, U]$*, Algorithm 1 is* $(1 + \ln(U/L))$*-competitive when the threshold is given by*

$$\phi(z) = \begin{cases} L & z \in [0, 1/(1 + \ln(U/L))) \\ L \exp((1 + \ln(U/L))z - 1) & z \in [1/(1 + \ln(U/L)), 1] \end{cases}. \tag{2}$$

*Further, no online algorithms can achieve a competitive ratio smaller than* $1 + \ln(U/L)$*.*

## 2.3 PREDICTION MODEL

We consider three prediction models for OFKP, each capturing a different prediction quality. All prediction models are constructed based on a critical value in the offline optimal solution. Thus,

we start by briefly describing the optimal offline solution for OFKP. Given that all item values and weights are known, the offline algorithm sorts the items in non-increasing order of unit value, and then greedily admits the sorted items until the knapsack capacity (See more detail in Appendix A.2).

**Definition 2.2** (Minimum acceptable items $(\hat{v}, \hat{\omega})$)**.** *Given an instance for OFKP, let $\hat{v}$ denote the minimum unit value of items admitted by the offline optimum, and $\hat{\omega}$ denote the total weights of items with the same unit value $\hat{v}$. Then $\hat{v}$ is defined as the critical value and $(\hat{v}, \hat{\omega})$ is defined as the minimum acceptable items for the instance.*

Below, we present three prediction models which consider how perfect or imperfect knowledge of $\hat{v}$ and $\hat{\omega}$ from Def. 2.2 allows us to recover the optimal solution.

**Prediction Model I** (Perfect Predictions)**.** *An exact point prediction of $\hat{v}$, as defined in Def. 2.2 (the minimum acceptable unit value), is given to the learning-augmented online decision maker.*

In the perfect prediction model, we assume that the learning-augmented decision maker has access to the exact minimum acceptable unit value $\hat{v}$ for any given instance $\mathcal{I} \in \Omega$. Note that the optimal offline solution will fully admit any item with a unit value strictly greater than $\hat{v}$. However, it may fractionally admit the item with value $\hat{v}$. Hence, even with a perfect prediction of exact value $\hat{v}$, the online decision-maker cannot optimally solve the problem since it is unclear how much of the item with value $\hat{v}$ should be admitted.

In practice, however, one may argue that it it almost impossible to obtain a perfect prediction of the exact value $\hat{v}$. Hence, we propose two extended prediction models which are practically relevant.

**Prediction Model II** (Interval Predictions)**.** *Deterministic lower and upper bounds on the actual value of $\hat{v}$ (Def. 2.2), are given to the learning-augmented online decision maker. Denote these by $\ell$ and $u$, respectively. Any such prediction satisfies $\ell \leq \hat{v} \leq u$.*

The second prediction model assumes that there is no exact prediction on $\hat{v}$. Instead, the predictions given to the algorithm are bounds on $\hat{v}$, i.e., $\hat{v} \in [\ell, u]$. The quality of prediction in this c $u - \ell$ increases. In an extreme case of $u = \ell$, the interval prediction degenerates to the aforementioned perfect prediction. On the other hand, with $u = U$ and $\ell = L$, the problem degenerates to the classic OFKP (see § 2.2) with prior knowledge on the unit value bounds.

**Prediction Model III** (Probabilistic Interval Predictions)**.** *As in Prediction Model II, the learning-augmented online decision maker receives lower and upper bounds $\ell$ and $u$ on $\hat{v}$ (Def. 2.2), and this prediction is probabilistically correct, i.e., $\mathbb{P}(\hat{v} \in [\ell, u]) = 1 - \delta$, where $\delta > 0$.*

Our last model relaxes the deterministic interval prediction into a probabilistic prediction such that with probability of $1 - \delta$, the critical value $\hat{v}$ is lower and upper bounded by $\ell$ and $u$. This relaxation allows us to analyze the case where predictions are untrusted and arbitrarily wrong.

In §3, we leverage the perfect prediction to design learning-augmented algorithms for OFKP and show the algorithms achieve the optimal competitive ratios. In §4, we study learning-augmented algorithms with imperfect prediction models and present an algorithm that utilizes the previously described algorithms as sub-algorithms to achieve practical competitiveness.

# 3 ALGORITHMS WITH PERFECT PREDICTION

## 3.1 LOWER BOUND RESULTS

We first present a lower bound result showing that even with a perfect prediction of the critical value $\hat{v}$, no deterministic or randomized learning-augmented algorithm can solve OFKP optimally.

**Theorem 3.1.** *Given an exact prediction on the critical value $\hat{v}$, no online algorithm for OFKP can achieve a competitive ratio smaller than $1 + \min\{1, \hat{\omega}\}$, where $\hat{\omega}$ is the total weight of items with the critical value for all instances in $\Omega$.*

The above result implies a lower bound of 2 on the competitive ratio, even if the algorithm has a perfect prediction of $\hat{v}$. Even if we know that the optimal solution admits an item with critical value $\hat{v}$, it is unclear how much weight of the knapsack should be filled with items of unit value $\hat{v}$.

## 3.2 PERFECT-PREDICTION-BASED ALGORITHMS

In this section, we first present PPA-n, a naïve perfect-prediction-based algorithm that can result in arbitrarily large competitive ratios. Then, we present PPA-b, a basic perfect-prediction-based algorithm for OFKP that achieves the competitive ratio of 2 given the exact prediction $\hat{v}$. Then, we present PPA-a, an advanced version of PPA-b that improves the competitive ratio to $1 + \hat{\omega}$, matching the lower bound value presented in Theorem 3.1. Recall that $\hat{\omega}$ is the weight of items with value $\hat{v}$ and *unknown* to all proposed algorithms.

**PPA-n:** *A naïve perfect-prediction-based algorithm:* We first consider a naïve "greedy" algorithm that takes a prediction on $\hat{v}$ as input. This algorithm rejects any items with unit value $< \hat{v}$ and fully accepts any item with unit value $\geq \hat{v}$ until the capacity limit. In Theorem 3.2, we show that PPA-n fails to achieve a meaningful improvement in the worst-case competitive ratio (i.e., consistency since we assume the prediction is correct).

**Theorem 3.2.** PPA-n *that fully trusts the prediction is* $U/L$*-competitive in the worst case.*

**PPA-b:** *A basic perfect-prediction-based 2-competitive algorithm.* We present an algorithm (Algorithm 2) that, given the exact prediction of $\hat{v}$, is 2-competitive for OFKP. The idea is to set aside half of the capacity only for high-value items (whose unit value $> \hat{v}$) and allocate the other half for admitting minimum acceptable items with exact value $\hat{v}$. By doing so, we will at least obtain half of either part in the optimal solution. Furthermore, this competitive ratio is optimal since no algorithm can achieve a competitive ratio smaller than 2 in the worst case as shown in Theorem 3.1.

---

**Algorithm 2** PPA-b: A basic 2-competitive algorithm for OFKP with perfect prediction

1: **input**: prediction $\hat{v}$.
2: **output**: online decisions $x_i$ s.
3: **while** item $i$ (with unit value $v_i$ and weight $w_i$) arrives **do**
4:     **if** $v_i < \hat{v}$ **then**
5:         $x_i = 0$;
6:     **else if** $v_i > \hat{v}$ **then**
7:         $x_i = w_i/2$;
8:     **else if** $v_i = \hat{v}$ **then**
9:         $x_i = \min(w_i/2, 1/2 - z)$;
10:         $z = z + x_i$;

---

**Theorem 3.3.** *Given a perfect prediction,* PPA-b *is* 2*-competitive.*

*Proof Sketch of Theorem 3.3.* Assuming unique prices, PPA-b selects half of each item with a unit value $v_i$ where $v_i \geq \hat{v}$. Essentially, it allocates half of the knapsack's capacity to items with values above $\hat{v}$ and the other half to items with values equal to $\hat{v}$. This intuitive approach gives us at least half the value of the offline optimal solution. The full proof is in Appendix A.3.3. □

We note that most existing works for OFKP and related problems (Marchetti-Spaccamela & Vercellis, 1995; El-Yaniv et al., 2001) make the assumption that item unit values are bounded, i.e., $v_i \in [L, U], \forall i \in [n]$, which is usually necessary to achieve non-trivial competitive bounds in the setting without predictions. Subsequent competitive bounds in these works depend on $U$, $L$, and the ratio between them (i.e., $U/L$). In our setting, the predictions allow us to achieve a constant competitive ratio that is independent of $U$ and $L$.

In the following, our goal is to propose a new algorithm that achieves a better parameterized competitive ratio than that of PPA-b. This can be accomplished by modifying Algorithm 2 to introduce a parameter, which represents the quantity we select when encountering $\hat{v}$ (instead of fixed $1/2$). This parameter allows us to reserve a portion for more valuable items more effectively.

Under the assumption that the unit value $v_i$ of each item in the instance is unique, (i.e. $i \neq j \rightarrow v_i \neq v_j \ \forall i, j \in [n]$), we can further refine the above concept to achieve a competitive ratio of $1 + \hat{\omega}$, where $\hat{\omega}$ denotes the weight of the *single item* with critical value $\hat{v}$. This modified algorithm can exhibit high efficiency, particularly when dealing with small values of $\hat{\omega}$. For example, in scenarios

---

**Algorithm 3** `PPA-a`: An advanced $(1 + \hat{\omega})$-competitive algorithm with perfect prediction

---

1: **input**: prediction $\hat{v}$.
2: **output**: online decisions $x_i$s.
3: b = 0, s = 0
4: **while** item $i$ (with unit value $v_i$ and weight $w_i$) arrives **do**
5:      **if** $v_i < \hat{v}$ **then**
6:         $x_i = 0;$
7:      **else if** $v_i > \hat{v}$ and $b = 0$ **then**
8:         $x_i = w_i$
9:         $s = s + x_i \times v_i$
10:      **else if** $v_i = \hat{v}$ **then**
11:         $b = 1$
12:         $x_i = \max\left((\hat{\omega}/(1 + \hat{\omega}) \times \hat{v} - s \times (\hat{\omega}/(1 + \hat{\omega})))/\hat{v}, 0\right)$
13:      **else if** $v_i > \hat{v}$ and $b = 1$ **then**
14:         $x_i = w_i/(\hat{\omega} + 1)$

---

resembling the $k$-search problem, where the weight of each item is $\frac{1}{k}$, this modification yields a competitive algorithm with a ratio of $1 + \frac{1}{k}$.

**`PPA-a`:** *An improved $(1 + \hat{\omega})$-competitive algorithm.* `PPA-a` leverages the observation that we can accept more than half of an item for values exceeding the prediction when $\hat{\omega}$ is low. However, since we don't have prior knowledge of $\hat{\omega}$, we exploit a "prebuying" strategy. Initially, we select all items with unit values $> \hat{v}$ and subsequently adjust our selections upon observing the prediction. This adaptive approach ensures that we select an appropriate portion of the prediction to achieve the desired competitive ratio. We note that here $\hat{\omega}$ represents the weight of a single item, since we assume that each item have a unique unit value.

**Theorem 3.4.** *Given perfect prediction,* `PPA-a` *achieves a competitive ratio of $1 + \hat{\omega}$ for* `OFKP` *with unique unit values.*

*Proof Sketch of Theorem 3.4.* Let's consider the scenario where we have knowledge of $\hat{\omega}$, and our goal is to achieve a competitive ratio of $1/(1 + \hat{\omega})$. One approach is to allocate $1/(1 + \hat{\omega})$ of the weight capacity for all items with $v_i > \hat{v}$ and an additional $1/(1 + \hat{\omega})$ for items with the same value as $\hat{v}$. This strategy intuitively yields a $(1 + \hat{\omega})$-competitive algorithm. We need to ensure that this allocation is feasible, which can be demonstrated to be the case.

If we do not have prior knowledge of $\hat{\omega}$, we can employ a "prebuying" strategy for all items with values higher than $\hat{v}$. The extra capacity allocated to these items each has a higher unit value than $\hat{v}$, allowing us to reduce our selection from $\hat{v}$ based on how much extra capacity we've allocated in previous items. The challenge here is to confirm the feasibility of this algorithm and provide a detailed analysis of its competitive ratio. The full proof is in Appendix A.3.4. □

## 4   ALGORITHMS WITH (PROBABILISTIC) INTERVAL PREDICTION

In this section, we further consider deterministic and probabilistic interval predictions that can model different levels of *imperfect predictions*. Our goal is to show that even with imperfect predictions, it is possible to devise learning-augmented algorithms with competitive ratios better than classic worst-case optimized algorithms that do not make use of any additional predictions.

### 4.1   DETERMINISTIC INTERVAL PREDICTION

We present an algorithm that uses a deterministic interval prediction $[\ell, u]$ for the critical value $\hat{v}$.
**`IPA`:** *An interval prediction-based algorithm.* `IPA` draws inspiration from `PPA-b` and devises an algorithm to solve `OFKP` with predictions represented as intervals. It allocates a dedicated portion of the capacity for values higher than $u$, and employs another algorithm, such as `TA`, to solve `OFKP` within the interval. The results are then combined to yield a competitive result with a competitive ratio of $\alpha + 1$, where $\alpha$ represents the competitive ratio of the sub-algorithm.

---

**Algorithm 4** `IPA`: An interval-prediction-based algorithm for `OFKP`

---

1: **input**: interval prediction $\ell, u$, robust algorithm $\mathcal{A}$ with competitive ratio $\alpha$
2: **Output**: Online decisions $x_i$s;
3: initialize $\mathcal{A}$
4: **while** item $i$ (with unit value $v_i$ and weight $w_i$) arrives **do**
5:     **if** $v_i < \ell$ **then**
6:         $x_i = 0$;
7:     **else if** $v_i > u$ **then**
8:         $x_i = 1/(\alpha + 1) \times w_i$;
9:     **else if** $v_i \in [\ell, u]$ **then**
10:         give item $i$ to algorithm $\mathcal{A}$ ;
11:         $x_i = \alpha/(\alpha + 1) \times x_i^{\mathcal{A}}$ ;

---

**Theorem 4.1.** *Given a deterministic interval prediction $[\ell, u]$ and a robust algorithm for* `OFKP` *with a competitive ratio of $\alpha$,* `IPA` *achieves a competitive ratio of $\alpha + 1$ for* `OFKP`.

*Proof Sketch of Theorem 4.1.* `IPA` resembles Algorithm 3. For unit values higher than $u$, this algorithm allocates $1/(\alpha + 1)$ of its weight. Within the range $[\ell, u]$, it employs a robust sub-algorithm, denoted as $\mathcal{A}$, which is $\alpha$-competitive. Using $\alpha/(\alpha + 1)$ of the results obtained from $\mathcal{A}$ intuitively yields a $(\alpha + 1)$-competitive solution for that range. The primary technical challenge is to demonstrate that we maintain competitiveness across all ranges. The full proof is in Appendix A.3.5. □

**Corollary 4.2.** `IPA` *is $2 + \ln(u/\ell)$-competitive for* `OFKP` *when the robust algorithm is given by Algorithm 1 (* `TA` *), for interval $[\ell, u]$.*

*Proof.* The result follows by observing that the `TA` algorithm presented in Algorithm 1 is $\ln(U/L) + 1$ competitive, where $U$ and $L$ are inputs to the algorithm. Letting $U = u$ and $L = \ell$, we have that $(\alpha + 1) = 2 + \ln(u/\ell)$. This competitive ratio will be close to 2 if $\ln(u/\ell)$ is small, and $\ln(u/\ell)$ becomes smaller as $\ell$ and $u$ approach each other. For $\ell = u$, we recover the same 2-competitive result as `PPA-b` (Algorithm 2). □

### 4.2 PROBABILISTIC INTERVAL PREDICTION

---

**Algorithm 5** `PIPA`: A probabilistic-interval-prediction-based algorithm of `OFKP`

---

1: **input**: $\gamma$, Prediction model $\mathcal{P}$, robust algorithm `TA` without predictions, algorithm $\mathcal{A}$ which uses $\mathcal{P}$ as input
2: **Output**: Online decisions $x_i$s.
3: initialize `TA` and $\mathcal{A}(\mathcal{P})$
4: **while** item $i$ (with unit value $v_i$ and weight $w_i$) arrives **do**
5:     give item $i$ to algorithm `TA`
6:     give item $i$ to algorithm $\mathcal{A}$
7:     $x_i = (1 - \gamma) \times x_i^{\text{TA}} + \gamma \times x_i^{\mathbf{A}}$;

---

**`PIPA`:** *A robust and consistent meta-algorithm.* `PIPA` deals with *imperfect predictions*, such as machine-learned predictions of $\hat{v}$ or the interval prediction $[\ell, u]$. This algorithm combines `TA`, the robust algorithm which uses no prediction and achieves competitive ratio $\ln(U/L) + 1$ (see Algorithm 1) with one of the prediction algorithms presented so far (`PPA-b` or `IPA`). If the prediction is correct, we say that the prediction algorithm is $c$-competitive.

For robustness purposes, we follow related work (Sun et al., 2021b; Marchetti-Spaccamela & Vercellis, 1995; El-Yaniv et al., 2001) and assume that item unit values are bounded, i.e., $v_i \in [L, U], \forall i \in [n]$. Note that $L$ and $U$ are not related to the predicted interval $[\ell, u]$. We balance between the sub-algorithms (`TA` and prediction `ALG`) by setting a *trust parameter* $\gamma \in [0, 1]$. Both algorithms run in parallel – when an item arrives, `PIPA` receives as input an item with unit value $v_i$, a weight $w_i$, and two decisions $\hat{x}_i$ and $\tilde{x}_i$, representing the decisions of the prediction and robust algorithms, respectively. Then `PIPA` simply purchases $x_i = \gamma \hat{x}_i + (1 - \gamma) \tilde{x}_i$ fraction of the item. Note that

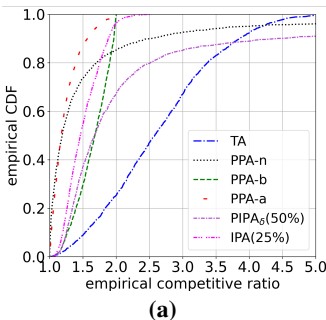 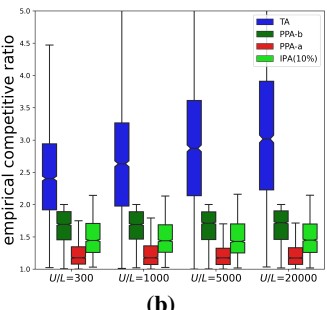 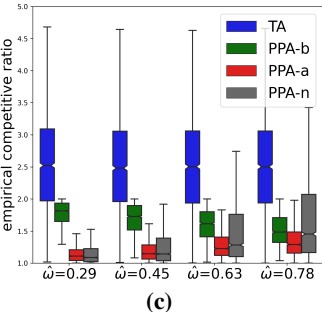

| (a) | (b) | (c) |

Figure 1: Performance comparison of different algorithms: **(a)** The CDF plot of the empirical competitive ratio of different algorithms; **(b)** The robust performance of `PPA-a`, `PPA-b` and `IPA` against `TA` when $U/L$ varies; and **(c)** The performance of `PPA-b`, `PPA-a`, and `PPA-n` against `TA` when $\hat{\omega}$ varies.

when $\gamma = 1$, `PIPA` will make the same decisions as the inner prediction algorithm, and when $\gamma = 0$, `PIPA` will make the same decisions as the inner robust algorithm `TA`. We assume that the inner prediction algorithm is chosen based on the type of prediction received, e.g. a point or interval prediction.

Suppose that the predictions are correct with probability $(1 - \delta) \in [0, 1]$. With $\delta = 1$, the predictions are always incorrect, and with $\delta = 0$, we recover the setting where the prediction is always correct. In Lemma 4.3, we give bounds on the consistency and robustness of this meta-algorithm.

**Lemma 4.3.** `PIPA` is $\frac{\ln(U/L)+1}{(1-\gamma)}$-robust and $\frac{c}{\gamma}$-consistent for any $\gamma \in (0, 1)$, where $c$ denotes the competitive ratio of the inner prediction algorithm with an accurate prediction.

*Proof Sketch of Lemma 4.3.* We first calculate the expected payoff of `PIPA` based on the trust parameter $\gamma$ and probability $\delta$ as $\mathbb{E}[\text{PIPA}[\gamma](\mathcal{I})] = \gamma \cdot (1 - \delta)\mathcal{A}(\mathcal{I}) + (1 - \gamma) \cdot \text{TA}(\mathcal{I})$. To analyze the consistency and robustness of `PIPA`, we consider two extreme cases for $\delta$ (i.e. when $\delta = 0$, the prediction is correct and we derive a consistency bound, and when $\delta = 1$, the prediction is always incorrect and we derive a robustness bound). The full proof is in Appendix A.3.6. □

As a corollary, Lemma 4.3 shows that the expected payoff of `PIPA` is $\gamma(1-\delta)\frac{\text{OPT}}{c} + (1-\gamma)\frac{\text{OPT}}{1+\ln(U/L)}$, even if $\delta$ is unknown. If the prediction has enough probability of being correct, e.g. $1 - \delta \geq \frac{c}{\ln(U/L)+1}$, increasing $\gamma$ will raise the expected payoff. Setting $\gamma$ close to 1 will result in a competitive algorithm with a ratio of approximately $c/(1-\delta)$. When using Algorithm 4, this competitiveness increases to approximately $2 + \ln(u/\ell)/(1 - \delta)$, which is particularly practical for small intervals.

## 5 NUMERICAL EXPERIMENTS

*Experimental setup and comparison algorithms.* To validate the performance of our algorithms, we conduct experiments using synthetically generated data, where the value and weight of items are randomly drawn from a power-law distribution. Unless otherwise mentioned, the lowest unit value is $L = 1$, and the highest unit value is $U = 1000$. Weights are drawn from a power law but normalized to be within the range of 0 to 1. We report the cumulative density functions of the empirical competitive ratios, which illustrate different algorithms' average and worst-case performances.

To report the empirical competitive ratio of different algorithms, we implement the offline optimal solution as described in Appendix A.2. We compare the results of the following online algorithms under various experimental settings: (1) `TA`: the classic online algorithms without prediction (Algorithm 1); (2) `PPA-n`: the naïve prediction-based algorithm; (3) `PPA-b`: the basic prefect-2-competitive algorithm (Algorithm 2); (4) `PPA-a`: the advanced $(1 + \hat{\omega})$-competitive algorithm (Algorithm 3); (5) `IPA`: the interval-prediction-based algorithm (Algorithm 4); and (6) probabilistic-interval-prediction-based (Algorithm 5). For the `IPA` algorithm, we present the interval prediction range $u - \ell$ as a percentage of range [L,U] and set it to three values of 15%, 25%, and 40%. For `PIPA`, the value of 1-$\delta$ is set to 10%, 20%, and 50%, and it is labeled as $\text{PIPA}_\delta$. Similarly, $\text{PIPA}_\gamma$ denotes that different variants `PIPA` under different values of the trust parameter $\gamma$, which in our

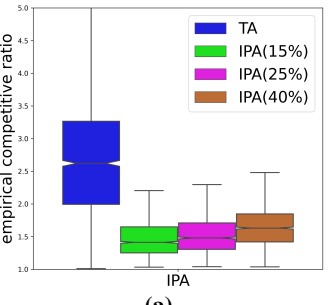 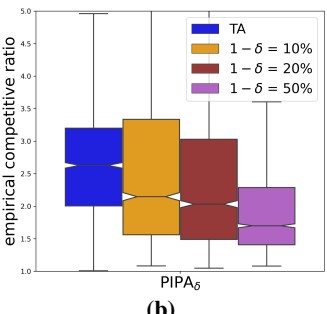 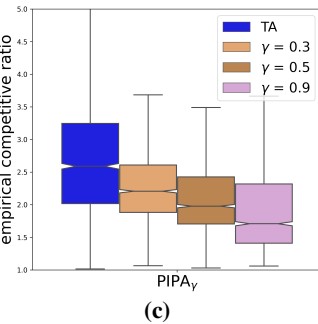

| (a) | (b) | (c) |

Figure 2: Interval-prediction-based algorithms with different interval sizes, probabilities $\delta$, and trust parameters $\gamma$: **(a)** Competitive ratios of three interval widths in `IPA` against baseline `TA`; **(b)** Competitive ratio of three probability $\delta$ values in `PIPA` against baseline `TA`. $\gamma = 0.9$ and interval 20%; and **(c)** Competitive ratio of three $\gamma$ values in `PIPA` against `TA`. $\delta = 50\%$ and interval 20%.

experiment we vary using three values of 0.3, 0.5, and 0.9. Furthermore, for `PPA-b`, $\hat{\omega}$ is set to four arbitrary values of 0.29, 0.45, 0.63, and 0.78.

*Experimental results.* Figure 1(a) reports the cumulative distribution function (CDF) of empirical competitive ratios for six different algorithms on 2000 synthetic instances of `OFKP`. The most notable observations are: **(1)** among all prediction-based algorithms, `PPA-a` achieves the best performance in both average and worst-case performance, verifying the theoretical results in Thm. 3.4. **(2)** while the average performance of `PPA-n` outperforms most algorithms (except `PPA-a`), its worst-case performance is even worse than `TA`, that do not leverage predictions in decision-making; this observation verifies the poor consistency of `PPA-n` reported in Thm. 3.2. **(3)** even with imperfect prediction, `IPA` outperforms `TA` in both average and worst-case results. **(4)** while achieving a bounded competitive ratio, `TA` performs on average worse than all prediction algorithms.

We now report the results of the impact of the parameters on the performance of different algorithms. First, our theoretical results show that different from classic algorithms such as `TA`, the competitive ratio of the prediction-based algorithms is independent of the ratio between the most and least valuable items, i.e., $U/L$. In Figure 1(b), we verify this theoretical observation by varying the value $U/L$ from 300 to 1000, 5000, and 20000. The results in Figure 1(b) show that the empirical competitive ratio of `TA` drastically increases as $U/L$ increases while other algorithms are robust to the variations of $U/L$. In Figure 1(c), we change the values of $\hat{\omega}$, and the results show that, in contrast to `PPA-n`, `PPA-b`, and `TA`, the performance of `PPA-a` substantially improves with smaller values of $\hat{\omega}$ verifying the results in Thm. 3.4 on the dependence of the competitive ratio of `PPA-a` on $\hat{\omega}$.

In Figure 2(a), we evaluate the performance of `IPA` for different interval prediction widths, given as a percentage (higher is worse). As shown in Theorem 4.1, we find that tighter prediction intervals yield better empirical performance. Furthermore, all `IPA` algorithms outperform the baseline robust `TA` algorithm. In Figure 2(b), we evaluate the performance of `PIPA` for *imperfect predictions*. We test regimes where $1 - \delta$ (probability of correct prediction) is 10%, 20%, and 50%; we fix $\gamma = 0.9$, and the interval is 20% of $[L, U]$. We find that the performance of `PIPA` smoothly degrades, and even bad predictions result in an algorithm that outperforms the robust baseline `TA`. Finally, in Figure 2(c), we show a similar result for `PIPA`– we fix $\delta = 50\%$ and vary the trust parameter $\gamma \in \{0.3, 0.5, 0.9\}$, showing that when predictions are sufficiently good, `PIPA` performs better when the predictions are trusted more (i.e., increasing $\gamma$). In Appendix A.4, we include additional results which further contextualize the performance of our proposed algorithms.

# 6    CONCLUSION

We study learning-augmented algorithms for the online fractional knapsack problem (`OFKP`) under predictions with varying quality. Given a perfect prediction, we have developed an online algorithm that can leverage the prediction and achieve the optimal competitive ratio. When the prediction is correct within an interval or probabilistically correct, we have further designed two algorithms that can use such imperfect predictions and achieve consistency and robustness improving average-case performance and giving worst-case guarantees, respectively. Through extensive numerical experiments, we validate that our proposed algorithms outperform all existing benchmark algorithms.

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

# A APPENDIX

## A.1 RELATED WORK

This paper contributes directly to two lines of work: (i) work on online knapsack, one-way trading, and related problems, e.g., $k$-search, single-leg revenue management, and; (ii) work on online algorithms with advice and learning-augmentation. We describe the relationship to each below.

**Online Knapsack and Online Search**   Our work contributes to the literature on the classic (integral) online knapsack problem first studied in (Marchetti-Spaccamela & Vercellis, 1995), with important results following in (Zhou et al., 2008). In the past few years, many works have considered variants of this problem, such as removable items (Cygan et al., 2016), item departures (Sun et al., 2022), and generalizations to multidimensional settings (Yang et al., 2021). Closer to this work, several studies have considered the online knapsack problem with additional information or in a learning-augmented setting, including frequency predictions (Im et al., 2021), online learning (Zeynali et al., 2021), and advice complexity (Böckenhauer et al., 2014b). However, unlike our work, most of these works do not generalize the problem to allow fractional item acceptance.

A few studies have considered online fractional knapsack under slightly different assumptions. Noga & Sarbua (2005) explore "online partially fractional knapsack", where items are removable. Giliberti & Karrenbauer (2021) consider online fractional knapsack in the random order model, where the arrival order of items is a random permutation (i.e. not adversarial). The setting in (Sun et al., 2021b) is perhaps the closest to the problem we study here, although it also considers a further generalization to the multiple knapsack problem. The study in (Sun et al., 2021b) motivates the observation that online fractional knapsack is equivalent to the one-way trading problem with a *rate constraint* for each online price.

The connection between OFKP and one-way trading (with a rate constraint) motivates a further connection to another track of literature on problems broadly classified as *online search*, including problems such as $1\text{-max}\,/\min$ search and one-way trading, both first studied by El-Yaniv et al. (2001), $k\text{-max}\,/\min$ search, first studied by Lorenz et al. (2008), and single-leg revenue management, first studied using competitive analysis by Ball & Queyranne (2009). In general, OFKP can be understood as a "bridge" between work on online search and online knapsack. Follow-up works have since considered applications of online search problems and additional variants, including cloud pricing (Zhang et al., 2017), electric vehicle charging (Sun et al., 2021b), switching cost of changing decisions (Lechowicz et al., 2023a), and learning-augmented versions of both one-way trading (Sun et al., 2021a) and $k$-search (Lee et al., 2022). However, to the best of our knowledge, none of these works consider the impact of rate constraints.

**Learning-augmented algorithms**   Learning-augmented algorithm design is an emerging field that incorporates machine-learned predictions about future inputs in algorithm designs, with the goal of matching the good average-case performance of the predictor while maintaining worst-case competitive guarantees. Introduced by (Lykouris & Vassilvtiskii, 2018; Purohit et al., 2018) in studies on caching and ski rental, respectively, the concepts of consistency and robustness give a formal mechanism to quantify the trade-off between following the machine learned predictions and hedging against adversarial inputs, particularly with respect to predictions which are very incorrect. This framework has been applied to a number of online problems, including caching Lykouris & Vassilvtiskii (2018), ski-rental (Purohit et al. (2018); Wei & Zhang (2020); Antoniadis et al. (2021)), set cover (Bamas et al. (2020)), online selection Jiang et al. (2021), online matching Antoniadis et al. (2020b), convex body chasing (Christianson et al. (2022)), and metrical task systems (Antoniadis et al. (2020a); Christianson et al. (2023)), just to name a few. Most relevant to our setting, it has been explored in the context of online knapsack (Im et al. (2021); Zeynali et al. (2021)), one-way trading (Sun et al. (2021a)), and single-leg revenue management (Balseiro et al. (2023)). However, to the best of our knowledge, learning-augmented algorithms have not been shown for the online fractional knapsack problem.

Learning-augmented algorithms are also closely related to the field of advice complexity, which considers how the performance of an online algorithm can be improved with a specific amount of advice about the input, where the advice is assumed to be correct and is provided by an oracle. This field was first established for paging by Böckenhauer et al. (2009), with results following

for many other online problems (Boyar et al. (2015); Böckenhauer et al. (2014a); Böckenhauer et al. (2011); Komm et al. (2012)). Of particular interest to our setting, Böckenhauer et al. (2014b) explore the online knapsack problem with advice, showing that a single bit of advice gives a 2-competitive algorithm, but that $\Omega(\log n)$ advice bits are necessary to further improve the competitive ratio. A similar result was also recently shown for the $k$-search problem by (Clemente et al. (2022)). However, to the best of our knowledge, there is no existing work considering the online fractional knapsack problem with advice.

### A.2  Offline Optimal Algorithm

The offline optimal solution to equation 1 () is straightforward to compute. The optimal solution starts by selecting the item with the maximum unit value amongst all $v_i$'s and adds the maximum amount allowed ($w_i$) to the knapsack. If there is any remaining capacity to fill, the optimal solution then picks the item with the next highest unit value and adds the maximum amount allowed while respecting the capacity constraint. This process is repeated until the knapsack is completely filled.

For ease of analysis, we let $(v'_i, w'_i)$ denote the values and weights of the items sorted in descending order. That is, $v'_1 \geq v'_2 \geq \ldots \geq v'_n$. We let $x'_i$ denote the portion of item $(v'_i, w'_i)$ which is added to the knapsack by some algorithm.

With this notation, the offline optimal solution to equation 1 can be written as:

$$\mathbf{x}^\star \overset{\text{def}}{=} (x_1^\star, \ldots, x_n^\star) = \left( w'_1, w'_2, \ldots, w'_{r-1}, 1 - \sum_{i=1}^{r-1} w'_i, 0, 0, \ldots \right). \tag{3}$$

As seen in equation 3, the optimal solution selects all the weight of the most valuable items until the knapsack capacity is filled. For lower values, it doesn't acceptance anything. We refer to the last item with a strictly positive acceptance as the $p$th item, which is the maximum value of $j \in [1, n]$ such that $\sum_{i=1}^{j-1} w'_i < 1$.

This optimal solution yields total profit:

$$\text{OPT}(\mathcal{I}) = \sum_{i=1}^{p-1} w'_i v'_i + \left( 1 - \sum_{i=1}^{p-1} w'_i \right) v'_p. \tag{4}$$

Both equation 3 and equation 4 hold if the sum of all $w_i$ is greater than or equal to 1, which constitutes the majority of interesting OFKP instances.

In cases where the sum of $w_i$ is less than 1, the optimal solution selects all items, which cannot be described by the above equations. To address this scenario, we introduce an auxiliary item denoted as $v_{n+1}$ with a corresponding weight $w_{n+1}$, where $v_{n+1} = 0$ and $w_{n+1} = 1$. It's important to note that this additional item doesn't affect the profit of any algorithm; rather, it simplifies and maintains consistency in mathematical modeling. In the problematic case, where the sum of $w_i$ is less than 1, $p$ would be equal to $n + 1$, and both equation 3 and equation 4 would still remain valid. We note that this case (where the knapsack can accept all items) is somewhat trivial, as the optimal policy simply accepts all items. In the majority of the paper, we implicitly assume that the sum of all $w_i$s is greater than 1.

### A.3  Proofs

#### A.3.1  Proof of Theorem 3.1

*Proof.* Consider the following two special instances in $\Omega$

$$\mathcal{I}_1: \quad n = 1, \quad (v_1, w_1) = (1, \hat{\omega}), \tag{5a}$$

$$\mathcal{I}_2: \quad n = 2, \quad (v_1, w_1) = (1, \hat{\omega}), \quad (v_2, w_2) = (U, 1 - \epsilon), \tag{5b}$$

where $U \to +\infty$ and $\epsilon \to 0$. The offline optimal returns of the two instances are $\text{OPT}(\mathcal{I}_1) = \hat{\omega}$ and $\text{OPT}(\mathcal{I}_2) = (1 - \epsilon)U + \epsilon$, respectively.

For both instances, the first item is the critical one and thus we are given the same prediction of the critical value 1. When we run an online algorithm over either instance, the algorithm makes the

same decision for the first item since the first item is the same and the prediction is the same. Let $X \in [0, \hat{\omega}]$ denote the decision for the first item from a randomized algorithm, where $X$ is a random variable following a probability density distribution $f(x)$. The expected returns of the randomized algorithm over the two instances can be derived as

$$\mathbb{E}[\text{ALG}(\mathcal{I}_1)] = \int_0^{\hat{\omega}} f(x)x\,dx = \mathbb{E}[X], \tag{6a}$$

$$\mathbb{E}[\text{ALG}(\mathcal{I}_2)] \leq \int_0^{\hat{\omega}} f(x)[x + (1-x)U]\,dx = \mathbb{E}[X] + (1 - \mathbb{E}[X])U, \tag{6b}$$

where $\mathbb{E}[X] \in [0, \hat{\omega}]$. As $U \to +\infty$ and $\epsilon \to 0$, the competitive ratio of the algorithm is

$$\max\left\{\frac{\text{OPT}(\mathcal{I}_1)}{\mathbb{E}[\text{ALG}(\mathcal{I}_1)]}, \frac{\text{OPT}(\mathcal{I}_2)}{\mathbb{E}[\text{ALG}(\mathcal{I}_2)]}\right\} \geq \max\left\{\frac{\hat{\omega}}{\mathbb{E}[X]}, \frac{1}{1 - \mathbb{E}[X]}\right\} \geq 1 + \hat{\omega}. \tag{7}$$

Thus, the competitive ratio of any online randomized algorithm is at least $1 + \hat{\omega}$. This completes the proof. $\qquad\square$

### A.3.2 PROOF OF THEOREM 3.2

*Proof.* Denote the prediction received by the algorithm as $\hat{v}$, for any valid OFKP instance $\mathcal{I}$.

Consider the following special instance in $\Omega$.

$$\mathcal{I}: \quad n = 2, \quad (v_1, w_1) = (L, 1), \quad (v_2, w_2) = (U, 1 - \epsilon), \tag{8a}$$

where $U \to +\infty$ and $\epsilon \to 0$. Note that the offline optimal return of this instance is $\text{OPT}(\mathcal{I}) = U(1 - \epsilon) + L(\epsilon)$, and $\hat{v} = L$.

Observe that the naïve algorithm will receive the exact value of $\hat{v}$ and greedily accept any items with unit value at or above $\hat{v}$. Then the first item with $(v_1, w_1) = (L, 1)$ will fill the online algorithm's knapsack, and the competitive ratio can be derived as

$$\frac{\text{OPT}(\mathcal{I})}{\text{ALG}(\mathcal{I})} = \frac{U(1 - \epsilon) + L(\epsilon)}{\hat{v}} = \frac{U(1 - \epsilon) + L(\epsilon)}{L}.$$

As $\epsilon \to 0$, the right-hand side implies that the competitive ratio is bounded by $U/L$. Since an accurate prediction has $\hat{v} \in [L, U]$ by definition, this special instance also gives the worst-case competitive ratio over all instances. $\qquad\square$

### A.3.3 PROOF OF THEOREM 3.3

*Proof.* Before starting the proof, we define a new notation for the optimal offline solution. Let's assume that there are $q - 1$ items with strictly greater values $v_i' > \hat{v}$ and items $q$ to $p - 1$ are items with unit value $\hat{v}$ ($q$ can equal $p - 1$, implying there are zero such items) . We can rewrite equation 4 as follows:

$$\text{OPT}(\mathcal{I}) = \sum_{i=1}^{q-1} w_i' v_i' + \sum_{i=q}^{p-1} w_i' \hat{v} + \left(1 - \sum_{i=1}^{p-1} w_i'\right) \hat{v}. \tag{9}$$

using the above notation we define $\hat{\omega}$ as:

$$\hat{\omega} := \sum_{i=q}^{r} w_i'. \tag{10}$$

where $r$ is largest number which $v_i' = \hat{v}$ and is greater or equal to $p$. Recall that $\hat{v} := v_p'$.

With this notation in place, we can proceed with the proof.

As described in Algorithm 2, each item $i$ with a value less than the prediction $\hat{v}$ is ignored. If the value is greater than the prediction, half of its weight is selected. If the prediction is equal to the

value, we will select half of it and ensure that the sum of all selections with a value equal to the prediction doesn't exceed $1/2$. We first show that Algorithm 2 outputs a feasible solution, i.e., that $\sum_{i=1}^{n} x_i \leq 1$. We derive the following equation:

$$\sum_{i=1}^{n} x_i = \left( \sum_{v_i < \hat{v}} x_i \right) + \left( \sum_{v_i = \hat{v}} x_i \right) + \left( \sum_{v_i > \hat{v}} x_i \right). \tag{11}$$

The first sum is equal to zero, since the algorithm doesn't select any items with $v_i < \hat{v}$. The second sum considers $v_i = \hat{v}$, and the algorithm selects half of every weight $w_i$ unless $\hat{\omega}$ is greater than $1/2$. The algorithm ensures it doesn't select more than $1/2$ by definition in line 9, which checks whether to take half of an item and not exceed the remaining amount from $1/2$. For $v_i' \geq v_p'$, Algorithm 2 sets $x_i$ to $w_i/2$.

$$\sum_{i=1}^{n} x_i = 0 + \min \left( \frac{\hat{\omega}}{2}, \frac{1}{2} \right) + \left( \sum_{v_i > \hat{v}} \frac{w_i}{2} \right). \tag{12}$$

The last sum is $1/2$ times the of the sum of $w_i$s for any items with a value greater than $\hat{v}$. If we look at equation 9, all of these $w_i$ are completely selected in the optimal offline solution (in the first part of the equation). Thus, their sum is less than or equal to 1, since the optimal solution is feasible: $\sum_{v_i > \hat{v}} w_i = \sum_{i=1}^{q-1} w_i' \leq 1$.

$$\sum_{i=1}^{n} x_i \leq \frac{1}{2} + \frac{1}{2} \cdot \left( \sum_{v_i > \hat{v}} w_i \right) \leq \frac{1}{2} + \frac{1}{2} = 1. \tag{13}$$

Equation 13 proves that the solution from Algorithm 2 is feasible.

We next calculate the profit obtained by Algorithm 2 and bound its competitive ratio. The profit can be calculated using equation 1 as:

$$\texttt{ALG}(\mathcal{I}) = \min \left( \frac{\hat{\omega}}{2}, \frac{1}{2} \right) \cdot \hat{v} + \left( \sum_{v_i > \hat{v}} \frac{w_i}{2} \cdot v_i \right). \tag{14}$$

Looking at equation 9, we claim that the second part of equation 14 is $\frac{1}{2} \cdot \sum_{i=1}^{q-1} w_i' v_i'$.

To calculate the competitive ratio, we give a bound on $\text{OPT}(\mathcal{I})/\text{ALG}(\mathcal{I})$ by substituting equation 9 and equation 14 into the definition of CR (i.e. $\texttt{OPT}/\texttt{ALG}$), obtaining the following:

$$\text{CR} = \max_{\mathcal{I} \in \Omega} \frac{\sum_{i=1}^{q-1} w_i' v_i' + \sum_{i=q}^{p-1} w_i' \hat{v} + \left( 1 - \sum_{i=1}^{p-1} w_i' \right) \hat{v}}{\sum_{i=1}^{q-1} \frac{w_i'}{2} v_i' + \min \left( \frac{\hat{\omega}}{2}, \frac{1}{2} \right) \cdot \hat{v}}, \tag{15}$$

$$\text{CR} = \max_{\mathcal{I} \in \Omega} \left( 2 \cdot \frac{\sum_{i=1}^{q-1} w_i' v_i' + \sum_{i=q}^{p-1} w_i' \hat{v} + \left( 1 - \sum_{i=1}^{p-1} w_i' \right) \hat{v}}{\sum_{i=1}^{q-1} w_i' v_i' + \min \left( \hat{\omega}, 1 \right) \cdot \hat{v}} \right). \tag{16}$$

Here we prove that the numerator is less than or equal to the denominator, which will give us equation 17.

$$\text{CR} = \max_{\mathcal{I} \in \Omega} \left( 2 \cdot \frac{\sum_{i=1}^{q-1} w_i' v_i' + \sum_{i=q}^{p-1} w_i' \hat{v} + \left( 1 - \sum_{i=1}^{p-1} w_i' \right) \hat{v}}{\sum_{i=1}^{q-1} w_i' v_i' + \min \left( \hat{\omega}, 1 \right) \cdot \hat{v}} \right) \leq 2. \tag{17}$$

We argue two cases: first, if $\hat{\omega} < 1$, then Algorithm 2 will always select half of every item with value $\hat{v}$. Then, by rewriting the value of $\hat{\omega}$ we have:

$$\sum_{i=1}^{q-1} w_i' v_i' + \min\left(\hat{\omega}, 1\right) \cdot \hat{v} = \sum_{i=1}^{q-1} w_i' v_i' + \left(\sum_{i=q}^{u} w_i'\right) \hat{v}$$

$$= \sum_{i=1}^{q-1} w_i' v_i' + \sum_{i=q}^{p-1} w_i' \hat{v} + w_p' \hat{v} + \sum_{i=p+1}^{u} w_i' \hat{v}. \tag{18}$$

Using the fact that the definition of $x_p^\star$ in equation 3 and LP constraint equation 1, we claim $1 - \sum_{i=1}^{p-1} w_i' = x_p^* \le w_p$. Thus, equation 18 is greater than the numerator in equation 17.

In the second case, if $\hat{\omega} \ge 1$, then the algorithm will stop selecting items with a value of $\hat{v}$ after it reaches a capacity of $1/2$ for those items. The optimal offline solution is feasible, so $\sum_{i=1}^{p-1} w_i' + 1 - (\sum_{i=1}^{p-1} w_i') \le 1$, which implies that the following is also true: $\sum_{i=q}^{p-1} w_i' + 1 - (\sum_{i=1}^{p-1} w_i') \le 1$. Using this, we can rewrite the numerator:

$$\sum_{i=1}^{q-1} w_i' v_i' + \sum_{i=q}^{p-1} w_i' \hat{v} + \left(1 - \sum_{i=1}^{p-1} w_i'\right) \hat{v} \le \sum_{i=1}^{q-1} w_i' v_i' + 1 \cdot \hat{v}. \tag{19}$$

Which subsequently implies that the numerator is less than or equal to the denominator.

Both cases have been proven, completing the proof of equation 17 – PPA-b is 2-competitive. □

### A.3.4    PROOF OF THEOREM 3.4

*Proof.* First, let's verify the correctness and feasibility of the algorithm. Assume that the $p'$-th item in the input instance was $\hat{v}$. If items are sorted in descending order, $w_p'$ is the weight of that item. Since items are considered to be unique, $\hat{\omega} = w_p' \le 1$. The sum of decision variables can be expressed as follows:

$$\sum_{i=1}^{n} x_i = \left(\sum_{v_i < \hat{v}} x_i\right) + x_p + \left(\sum_{v_i > \hat{v}} x_i\right). \tag{20}$$

The third sum can be further rewritten as the sum of items before $p'$ and after $p'$:

$$\left(\sum_{v_i > \hat{v}} x_i\right) = \left(\sum_{v_i > \hat{v}, i < p'} x_i\right) + \left(\sum_{v_i > \hat{v}, i > p'} x_i\right). \tag{21}$$

Using the definition of the algorithm to calculate each part, the result is as follows:

$$= 0 + \max\left(\left(\left(\frac{\hat{\omega}}{1+\hat{\omega}} \cdot \hat{v} - s \cdot \frac{\hat{\omega}}{1+\hat{\omega}}\right) \cdot \frac{1}{\hat{v}}, 0\right)\right) + \left(\sum_{v_i > \hat{v}, i < p'} w_i\right) + \left(\sum_{v_i > \hat{v}, i > p'} \frac{w_i}{1+\hat{\omega}}\right). \tag{22}$$

Where $s$ is (by definition in the algorithm) the sum of the profit of all items with $v_i > \hat{v}$ that came before $\hat{v}$, formalized as $\sum_{v_i > \hat{v}, i < p'} w_i \cdot v_i$. Now, let us consider two cases for the maximum term in the above equation.

**I.** In case the left part of the maximum is negative, we have:

$$\sum_{i=1}^{n} x_i = \left(\sum_{v_i > \hat{v}, i < p'} w_i\right) + \left(\sum_{v_i > \hat{v}, i > p'} \frac{w_i}{1+\hat{\omega}}\right) < \left(\sum_{v_i > \hat{v}} w_i\right) < 1. \tag{23}$$

This is clearly less than 1 since only values higher than the critical value are used, and the sum of their $w_i$ is less than 1, as referenced in equation 3.

**II.** In case the left part of the maximum is positive, we have:

$$\sum_{i=1}^{n} x_i = \frac{\hat{\omega}}{1 + \hat{\omega}}$$

$$- \left( \sum_{v_i > \hat{v}, i < p'} w_i \cdot v_i \right) \cdot \frac{\hat{\omega}}{1 + \hat{\omega}} \cdot \frac{1}{\hat{v}} \tag{24}$$

$$+ \left( \sum_{v_i > \hat{v}, i < p'} w_i \right) + \left( \sum_{v_i > \hat{v}, i > p'} \frac{w_i}{1 + \hat{\omega}} \right).$$

After simplification, we obtain:

$$\sum_{i=1}^{n} x_i = \frac{\hat{\omega}}{1 + \hat{\omega}} + \left( \sum_{v_i > \hat{v}, i < p'} w_i \cdot \left( 1 - \frac{\hat{\omega}}{1 + \hat{\omega}} \cdot \frac{v_i}{\hat{v}} \right) \right) + \left( \sum_{v_i > \hat{v}, i > p'} \frac{w_i}{1 + \hat{\omega}} \right). \tag{25}$$

Now it can be observed that since $v_i > \hat{v}$, this results in the following inequality:

$$\sum_{i=1}^{n} x_i < \frac{\hat{\omega}}{1 + \hat{\omega}} + \left( \sum_{v_i > \hat{v}, i < p'} w_i \cdot \left( 1 - \frac{\hat{\omega}}{1 + \hat{\omega}} \right) \right) + \left( \sum_{v_i > \hat{v}, i > p'} \frac{w_i}{1 + \hat{\omega}} \right). \tag{26}$$

Furthermore, it follows that the left sum entries are also equal to $w_i/(1 + \hat{\omega})$:

$$\sum_{i=1}^{n} x_i < \frac{\hat{\omega}}{1 + \hat{\omega}} + \left( \sum_{v_i > \hat{v}} \frac{w_i}{1 + \hat{\omega}} \right). \tag{27}$$

Since the sum of $w_i$ for items with higher unit values is less than 1, as shown in equation 3, it can be concluded that:

$$\sum_{i=1}^{n} x_i < \frac{\hat{\omega}}{1 + \hat{\omega}} + \frac{1}{1 + \hat{\omega}} = 1. \tag{28}$$

This proves that the solution is feasible.

Now, we prove that the algorithm achieves a competitive ratio of $1 + \hat{\omega}$. The profit can be calculated in a straightforward manner based on $x_i$ decisions using equation 20, equation 21, and equation 22:

$$\text{ALG}(\mathcal{I}) = \max \left( \left( \frac{\hat{\omega}}{1 + \hat{\omega}} \cdot \hat{v} - s \cdot \frac{\hat{\omega}}{1 + \hat{\omega}} \right), 0 \right)$$

$$+ \left( \sum_{v_i > \hat{v}, i < p'} w_i \cdot v_i \right) + \left( \sum_{v_i > \hat{v}, i > p'} \frac{w_i}{1 + \hat{\omega}} \cdot v_i \right). \tag{29}$$

Let us consider the same two cases for the maximum.
**I.** In case the left part of the maximum is negative, we have:

$$\text{ALG}(\mathcal{I}) = \left( \sum_{v_i > \hat{v}, i < p'} w_i \cdot v_i \right) + \left( \sum_{v_i > \hat{v}, i > p'} \frac{w_i}{1 + \hat{\omega}} \cdot v_i \right). \tag{30}$$

which can be rewritten as:

$$\text{ALG}(\mathcal{I}) = \left( \sum_{v_i > \hat{v}, i < p'} \frac{w_i \cdot \hat{\omega}}{1 + \hat{\omega}} \cdot v_i \right) + \left( \sum_{v_i > \hat{v}, i < p'} \frac{w_i}{1 + \hat{\omega}} \cdot v_i \right) + \left( \sum_{v_i > \hat{v}, i > p'} \frac{w_i}{1 + \hat{\omega}} \cdot v_i \right). \tag{31}$$

Using the definition of $s$, we can rewrite the left sum as:

$$\text{ALG}(\mathcal{I}) = s \cdot \frac{\hat{\omega}}{1 + \hat{\omega}} + \left( \sum_{v_i > \hat{v}, i < p'} \frac{w_i}{1 + \hat{\omega}} \cdot v_i \right) + \left( \sum_{v_i > \hat{v}, i > p'} \frac{w_i}{1 + \hat{\omega}} \cdot v_i \right). \tag{32}$$

Since we know that $s \cdot \frac{\hat{\omega}}{1+\hat{\omega}} > \frac{\hat{\omega}}{1+\hat{\omega}} \cdot \hat{v}$, we conclude that:

$$\text{ALG}(\mathcal{I}) > \frac{\hat{\omega}}{1+\hat{\omega}} \cdot \hat{v} + \left( \sum_{v_i > \hat{v}, i < p'} \frac{w_i}{1+\hat{\omega}} \cdot v_i \right) + \left( \sum_{v_i > \hat{v}, i > p'} \frac{w_i}{1+\hat{\omega}} \cdot v_i \right). \tag{33}$$

And a further simplification of equation 33 gives:

$$\text{ALG}(\mathcal{I}) > \frac{1}{1+\hat{\omega}} \cdot \left( \hat{\omega} \cdot \hat{v} + \left( \sum_{v_i > \hat{v}} w_i \cdot v_i \right) \right). \tag{34}$$

Since we can say that $\hat{\omega} \geq 1 - \sum_{i < p'} w_i'$, we claim that:

$$\text{ALG}(\mathcal{I}) > \frac{1}{1+\hat{\omega}} \cdot \left( 1 - \sum_{i < p'} w_i' \cdot \hat{v} + \left( \sum_{v_i > \hat{v}} w_i \cdot v_i \right) \right) = \frac{1}{1+\hat{\omega}} \text{OPT}(\mathcal{I}), \tag{35}$$

which implies that the competitive ratio is $1 + \hat{\omega}$.

**II.** In case the left part of the maximum is positive, we have:

$$\text{ALG}(\mathcal{I}) = \left( \frac{\hat{\omega}}{1+\hat{\omega}} \cdot \hat{v} - s \cdot \frac{\hat{\omega}}{1+\hat{\omega}} \right) + \left( \sum_{v_i > \hat{v}, i < p'} w_i \cdot v_i \right) + \left( \sum_{v_i > \hat{v}, i > p'} \frac{w_i}{1+\hat{\omega}} \cdot v_i \right). \tag{36}$$

Rewriting the middle sum as in equation 31 and equation 32 results in:

$$\begin{aligned}
\text{ALG}(\mathcal{I}) = &\left( \frac{\hat{\omega}}{1+\hat{\omega}} \cdot \hat{v} - s \cdot \frac{\hat{\omega}}{1+\hat{\omega}} \right) \\
&+ s \cdot \frac{\hat{\omega}}{1+\hat{\omega}} + \left( \sum_{v_i > \hat{v}, i < p'} \frac{w_i}{1+\hat{\omega}} \cdot v_i \right) + \left( \sum_{v_i > \hat{v}, i > p'} \frac{w_i}{1+\hat{\omega}} \cdot v_i \right).
\end{aligned} \tag{37}$$

This equation is identical to equation 33. Therefore, the same bound is achieved, and this equation is greater than $\frac{w_i}{1+\hat{\omega}} \text{OPT}(\mathcal{I})$, which confirms that the competitive ratio is $1 + \hat{\omega}$.

$\square$

### A.3.5 PROOF OF THEOREM 4.1

*Proof.* First, we analyze the feasibility of the solution – we show that $\sum_{i=1}^{n} x_i \leq 1$.

$$\sum_{i=1}^{n} x_i = \left( \sum_{v_i < \ell} x_i \right) + \left( \sum_{v_i \in [\ell, u]} x_i \right) + \left( \sum_{v_i > u} x_i \right). \tag{38}$$

By substituting sub-algorithm selections, we have the following:

$$\sum_{i=1}^{n} x_i = 0 + \left( \sum_{v_i \in [\ell, u]} \frac{\alpha}{\alpha+1} \cdot x_i^A \right) + \left( \sum_{v_i > u} \frac{1}{\alpha+1} \cdot w_i \right). \tag{39}$$

Using the fact that the sub-algorithm is also a feasible algorithm, we can say that:

$$\left( \sum_{v_i \in [\ell, u]} \frac{\alpha}{\alpha+1} \cdot x_i^A \right) = \frac{\alpha}{\alpha+1} \cdot \left( \sum_{v_i \in [\ell, u]} x_i^A \right) \leq \frac{\alpha}{\alpha+1} \cdot 1. \tag{40}$$

Also, from equation 3, we know that the optimal solution will select every $w_i$ with $v_i > \hat{v}$, since $\hat{v} \in [\ell, u]$. We can say that $w_i$ for which $v_i > u$ implies $v_i > \hat{v}$. Moreover, equation 4 is a feasible result, we know that $\sum_{v_i > \hat{v}} w_i \leq 1$. So, we claim that:

$$\left( \sum_{v_i > u} \frac{1}{\alpha + 1} \cdot w_i \right) = \frac{1}{\alpha + 1} \cdot \left( \sum_{v_i > u} w_i \right) \leq \frac{1}{\alpha + 1} \cdot \left( \sum_{v_i > \hat{v}} w_i \right) \leq \frac{1}{\alpha + 1} \cdot 1. \tag{41}$$

Substituting equation 41 and equation 40 into equation 39, we obtain:

$$\sum_{i=1}^{n} x_i = 0 + \left( \sum_{v_i \in [\ell, u]} \frac{\alpha}{\alpha + 1} \cdot x_i^A \right) + \left( \sum_{v_i > u} \frac{1}{\alpha + 1} \cdot w_i \right) \leq \frac{\alpha}{\alpha + 1} + \frac{1}{\alpha + 1} = 1, \tag{42}$$

which completes the proof that the solutions are feasible.

We proceed to prove that the algorithm achieves a competitive ratio of $\alpha + 1$ (given sub-algorithm TA, which has a competitive ratio $\alpha$). The profit can be calculated based on $x_i$ decisions, using equation 39:

$$\text{ALG}(\mathcal{I}) = \left( \sum_{v_i \in [\ell, u]} \frac{\alpha}{\alpha + 1} \cdot x_i^A \cdot v_i \right) + \left( \sum_{v_i > u} \frac{1}{\alpha + 1} \cdot w_i \cdot v_i \right). \tag{43}$$

For the first sum, we know that Algorithm TA guarantees $\alpha$-competitiveness, which we show as follows:

$$\left( \sum_{v_i \in [\ell, u]} x_i^A \cdot v_i \right) \times \alpha \geq \text{OPT}(\mathcal{I}_{\prime}), \tag{44}$$

where $\mathcal{I}_{\prime}$ is all items in $\mathcal{I}$ such that $v_i \in [\ell, u]$. Also, we claim that:

$$\text{OPT}(\mathcal{I}_{\prime}) = \sum_{u > v_i' > \ell, i < p'} w_i' \cdot v_i' + \left( 1 - \sum_{u > v_i' > \ell, i < p'} w_i' \right) \cdot v_p^{I_0}. \tag{45}$$

Where $v_p^{I_0}$ represents the critical value for $\mathcal{I}_{\prime}$. We argue that $v_p^{I_0} \leq \hat{v}$. If we denote $\hat{v}$ as the $p$th item in the sorted list, as defined in equation 4, and $v_p^{I_0}$ as the $p'$th item in the sorted list of the instance $\mathcal{I}$, we assert that $p \leq p'$. The rationale behind this assertion is rooted in the definitions. Specifically, $p$ is defined as the largest number for which $\sum_{i=1}^{p-1} w_i' < 1$. Now, let us assume $k$ is the first item with a value less than or equal to $u$. By definition, $p'$ represents the largest number for which $\sum_{i=k}^{p'-1} w_i' < 1$. If we were to assume that $p > p'$, this would contradict the definition of $p'$ as the largest number, because changing $p'$ to $p$ would yield a sum less than one, but we increased from the $p'$ to another larger number. Consequently, it is not valid to claim that $p > p'$; instead, we conclude that $p \leq p'$. This implies $v_p^{I_0} \leq \hat{v}$.

Using this observation, we can now compare $\text{OPT}(\mathcal{I})$ and $\text{OPT}(\mathcal{I}_I)$:

$$
\begin{aligned}
\text{OPT}(\mathcal{I}) &= \sum_{i=1}^{p-1} w'_i v'_i + \left(1 - \sum_{i=1}^{p-1} w'_i\right) \hat{v}, \\
&= \sum_{v_i > u} w'_i v'_i + \sum_{i=k}^{p-1} w'_i v'_i + \left(1 - \sum_{i=1}^{p-1} w'_i\right) \hat{v}, \\
&\leq \sum_{v_i > u} w'_i v'_i + \sum_{i=k}^{p-1} w'_i v'_i + \sum_{i=p}^{p'-1} w'_i v'_i + \left(1 - \sum_{i=k}^{p'-1} w'_i\right) v_p^{I_0}, \\
&= \sum_{v_i > u} w_i v_i + \text{OPT}(\mathcal{I}_I),
\end{aligned}
\tag{46}
$$

where equation 46 holds because if $p = p'$, then we have $\hat{v} = v_p^{I_0}$, implying that the last sum of $\text{OPT}(\mathcal{I})$ (which is $\left(1 - \sum_{i=1}^{p-1} w'_i\right) \hat{v}$), is equal to $\left(\sum_{i=k}^{p'-1} w'_i\right) v_p^{I_0}$.

On the other hand, if $p < p'$, the last part of $\text{OPT}(\mathcal{I})$ can be bounded by $w'_p v'_p$, which is subsumed within $\sum_{i=p}^{p'-1} w'_i v'_i$. This follows from the fact that $1 - \sum_{i=1}^{p-1} w'_i < w'_p$ due to the definition of a feasible answer.

Using equation 44 and equation 46, we claim that:

$$
\left(\sum_{v_i \in [\ell, u]} x_i^A \cdot v_i\right) \geq \frac{1}{\alpha} \cdot \left(\text{OPT}(\mathcal{I}) - \sum_{v_i > u} w_i v_i\right).
\tag{47}
$$

Now, let us combine this with other parts in equation 43:

$$
\begin{aligned}
\text{ALG}(\mathcal{I}) &= \left(\sum_{v_i \in [\ell, u]} \frac{\alpha}{\alpha + 1} \cdot x_i^A \cdot v_i\right) + \left(\sum_{v_i > u} \frac{1}{\alpha + 1} \cdot w_i \cdot v_i\right) \\
&\geq \left(\frac{\alpha}{\alpha + 1} \cdot \frac{1}{\alpha}\left(\text{OPT}(\mathcal{I}) - \sum_{v_i > u} w_i v_i\right)\right) + \left(\frac{1}{\alpha + 1} \cdot \sum_{v_i > u} w_i \cdot v_i\right) \\
&= \frac{1}{\alpha + 1} \text{OPT}(\mathcal{I}).
\end{aligned}
\tag{48}
$$

Thus, we conclude that $\text{IPA}$ using $\text{TA}$ as the robust sub-algorithm is $\alpha + 1$ competitive. $\qquad \square$

### A.3.6  Proof of Lemma 4.3

*Proof.* First, we denote the payoff of the solutions obtained by the prediction sub-algorithm and the robust sub-algorithm by $\mathcal{A}$ and $\text{TA}$, respectively.

Given that the prediction is correct with probability $(1 - \delta)$, we derive the expected payoff of $\text{PIPA}$ on an arbitrary instance $\mathcal{I}$ as follows:

$$
\begin{aligned}
\mathbb{E}[\text{PIPA}[\gamma](\mathcal{I})] &= \gamma \cdot (1 - \delta)\mathcal{A}(\mathcal{I}) + (1 - \gamma) \cdot \text{TA}(\mathcal{I}), \\
\mathbb{E}[\text{PIPA}[\gamma](\mathcal{I})] &= \gamma \cdot (1 - \delta)\frac{\text{OPT}(\mathcal{I})}{c} + (1 - \gamma) \cdot \frac{\text{OPT}(\mathcal{I})}{1 + \ln(U/L)}.
\end{aligned}
$$

To analyze the consistency and robustness of $\text{PIPA}$, we then consider two extreme cases for $\delta$.

If $\delta = 1$, the prediction is always incorrect and we consider the *robustness* of the meta-algorithm as follows:

$$\mathbb{E}[\mathrm{PIPA}[\gamma](\mathcal{I})] \geq (1 - \gamma)\frac{\mathrm{OPT}(\mathcal{I})}{1 + \ln(U/L)},$$

$$\frac{\mathrm{OPT}(\mathcal{I})}{\mathbb{E}[\mathrm{PIPA}[\gamma](\mathcal{I})]} \leq \frac{1 + \ln(U/L)}{(1 - \gamma)}.$$

On the other hand, if $\delta = 0$, the prediction is always correct and we consider the *consistency* of the meta-algorithm as follows.

$$\mathbb{E}[\mathrm{PIPA}[\gamma](\mathcal{I})] \geq \gamma\frac{\mathrm{OPT}(\mathcal{I})}{c} + (1 - \gamma) \cdot \frac{\mathrm{OPT}(\mathcal{I})}{1 + \ln(U/L)},$$

$$\geq \gamma\frac{\mathrm{OPT}(\mathcal{I})}{c},$$

$$\frac{\mathrm{OPT}(\mathcal{I})}{\mathbb{E}[\mathrm{PIPA}[\gamma](\mathcal{I})]} \leq \frac{c}{\gamma}.$$

$\square$

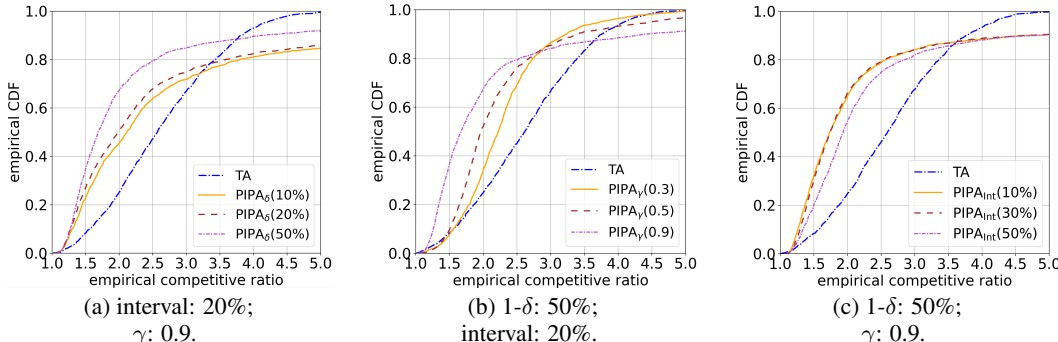

Figure A1: Performance of the probabilistic-interval-prediction-based algorithm (`PIPA`) while varying parameters (probability of correct prediction $(1-\delta)$, trust parameter $\gamma$, and interval size) against the robust online threshold-based algorithm (`TA`).

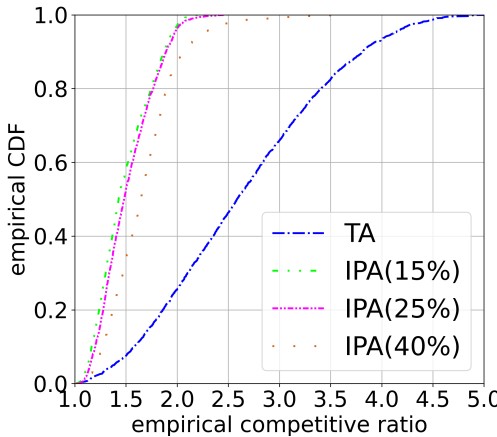

Figure A2: The performance of interval-prediction-based algorithm (`IPA`) with three intervals against online threshold-based algorithm (`TA`).

Figure A3: The Performance of probabilistic-interval-prediction-based algorithm (`PIPA`) with different intervals against online threshold-based algorithm (`TA`). $\delta = 50\%, \gamma = 0.9$

## A.4 ADDITIONAL NUMERICAL EXPERIMENTS

In this section, we report additional experimental results of the performance of the proposed algorithms.

In Figure A1(a), we evaluate the performance of `PIPA` for different values of $1 - \delta$. This plot is a CDF plot version of Figure 2(b), which illustrates how increasing the probability of correct predictions (corresponding to empirically more accurate machine-learned predictions), we consistently achieve a better competitive ratio, both on average and in the worst case.

In the second part, Figure A1(b), we examine the trust parameter $\gamma$ with values $0.3, 0.5, 0.9$ for `PIPA`. This figure represents a CDF plot of Figure 2(c) and shows that as the algorithm increases its trust in predictions (i.e. by increasing $\gamma$), the average competitive ratio improves. However, the worst-case competitive ratio (represented in this plot by the *tail* of the CDF) will deteriorate faster when placing more trust in predictions.

In Figure A2, which is a CDF plot of Figure 2(a), we evaluate the empirical competitive ratio of `IPA` for various interval prediction widths, represented as a percentage (where higher values indicate worse performance). We test values of $15\%$, $25\%$, and $40\%$ of the "width" of the interval $[L, U]$. Intuitively, reducing the interval size generally results in a better competitive ratio, because the upper and lower bounds on the value of $\hat{v}$ are tighter.

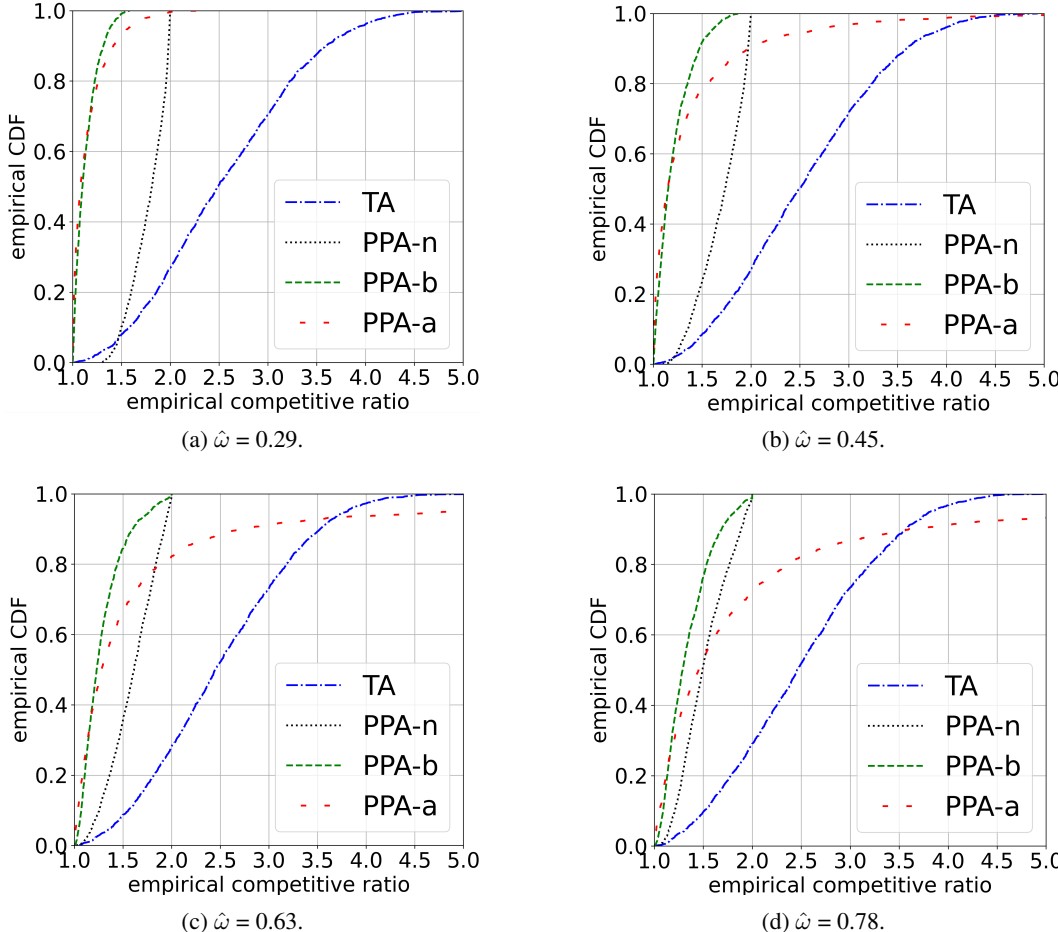

Figure A4: The performance of naïve greedy algorithm (`PPA-n`), basic 2-competitive algorithm (`PPA-b`), and advanced $(1+\hat{\omega})$-competitive algorithm (`PPA-a`) against threshold-based algorithm (`TA`) with varying $\hat{\omega}$.

Similarly, Figure A1(c) demonstrates that the width of the prediction interval (tested as 10%, 30%, and 50% of the "width" of the interval $[L, U]$) also has an slight effect on the average and worst-case competitive ratio for `PIPA`– namely, tighter prediction intervals yield better empirical performance, which aligns with our expectations.

Figure A3 is a box plot version of Figure 2(c), illustrating that `PIPA`'s average-case performance is not significantly impacted by the width of the predicted interval, although tighter intervals are still intuitively better.

Finally, in Figure A4, we vary the actual tested value of $\hat{\omega}$ in four CDF plots, one for each tested value $(0.29, 0.45, 0.63,$ and $0.78)$. In contrast to `PPA-n`, `PPA-b`, and `TA`, these results show that the performance of `PPA-a` substantially improves with smaller values of $\hat{\omega}$, confirming the results in Theorem 3.4, which establish that `PPA-a`'s competitive ratio depends on $\hat{\omega}$. These figures correspond to the CDF plot of Figure 1(c). Another interesting observation is that as $\hat{\omega}$ decreases, `PPA-b`'s empirical performance worsens, but for high $\hat{\omega}$ values, it improves. This occurs because `PPA-b` is designed to target 2-competitiveness, but performs well when selecting more items in $\hat{v}$ than the optimal solution. This occurs when $\hat{\omega}$ is high, as `PPA-b` chooses half of it.

