# OpenReview forum: "Online Fractional Knapsack With Predictions"
_ICLR.cc/2024/Conference — Submitted to ICLR 2024_

### Official Review · Reviewer_G3xe · 2023-10-31

**Soundness:** 3 good
**Presentation:** 3 good
**Contribution:** 2 fair
**Rating:** 3
**Confidence:** 4

**Summary:**

This document discusses the online fractional knapsack problem, where items can be fractionally accepted into a capacity-limited knapsack. The authors present online algorithms for this problem that incorporate predictions about the input, including predictions of the smallest value chosen in the optimal offline solution and interval predictions that give upper and lower bounds on this smallest value. They prove the competitive ratios of these algorithms and provide a matching worst-case lower bound. Additionally, they introduce a learning-augmented meta-algorithm that combines prediction techniques with a robust baseline algorithm to achieve reasonable consistency and robustness. The authors conduct numerical experiments that demonstrate the superiority of their prediction algorithms compared to a simple greedy prediction algorithm and the baseline algorithm. They also show that imperfect predictions can greatly improve average-case performance without sacrificing worst-case guarantees.

**Strengths:**

1.	It is interesting to consider predictions in the online fractional knapsack problem.
2.	Based on the prefect prediction of the smallest value in the optimal offline solution, the paper provides matched upper bound and lower bound of the competitive ratio.
3.	The intuition of proof is clear, and it is convenient to understand for the reader.

**Weaknesses:**

1. Random order model is mostly studied in the online (fractional) knapsack problem, since the adversary model is too strong to obtain good results. In the adversary model, we must know some prior information about the input in advance, otherwise it is impossible to expect reasonable competitive ratio. In other hands, fractional knapsack problem is often studied as a relax of knapsack problem. Therefore, the model to consider online fractional knapsack problem under adversary model is in some sense artificial.  What was worse, the prediction model in the paper requires the algorithm to predict the smallest unit value in the optimal solution. This is not natural since the algorithm does not just know the information of the input (like the interval of all unit value of items), but also know some information of the optimal solution. The paper does not explain the motivation of the model setting.
2. The technology used in the paper is relatively simple, it is just a simple linear combination of two models, and there is not much technical innovation.
3. The paper contains complex mathematical equations and notations that may be difficult for readers to understand. The lack of clear explanations and simplifications could make it challenging for readers to grasp the concepts and follow the arguments presented. For example, the A algorithm in Algorithm 4 and Algorithm 5 does not refer to the same algorithm, which will confuse readers.

**Questions:**

1.	Is there any motivation to study the online fractional knapsack problem based on adversary model and the prediction model discussed in the paper?
2.	The related work of the online knapsack problem with prediction is not very clear. Is it possible to compare the results of the online knapsack problem with prediction and the online fractional knapsack problem with prediction from different perspectives?

---

> ### Author Response · Authors · 2023-11-16
>
> Thank you for your review and suggestions. We will take them into account while revising the paper. Below we answer your questions in detail:
>
> - **(motivation for fractional knapsack problem)**
> While it is true that online fractional knapsack (OFKP) doesn't address all online knapsack problems, there are numerous problems that can be effectively modeled by OFKP. This includes, for example, dynamic bandwidth allocation in wireless or communication networks [3, 4], online electric vehicle charging in smart grids [5], and online trading problems in financial markets [5]. For bandwidth allocation problems, multiple users share the same bandwidth with fixed capacity. Upon the arrival of each user, it requests for at most $w_i$ amount of bandwidth and each unit of bandwidth has a value $v_i$. The system operator determines how much bandwidth is allocated to the user with the objective of maximizing the total profit of the allocated bandwidth. OFKP exactly models this problem and the integral version of the online knapsack cannot provide a meaningful solution, especially when $w_i$ is comparable to the bandwidth capacity.
> We do agree that it would be interesting to extend our approach to integral knapsack problems – often results for the fractional case can be extended to the integral case if we assume that the item sizes are small enough. However, detailed derivation of the results needs further analysis which is beyond the scope of current work.
> - **(random order model vs adversarial model)**
> In our work, we focus on the most general adversarial model of input. As you point out, many prior works assume additional structure on the input to obtain better results (e.g., random order model or stochastic input model). However, the resulting algorithms and performance bounds for the problem with these assumptions are quite different from the adversarial model.  Since the adversarial model does not assume any structure on the input, it is strictly more general than e.g. the random order model, which is a strength of our work. By simultaneously considering adversarial inputs and (possibly untrusted or imperfect) predictions about the input, we seek to obtain the “best of both worlds” between the fully adversarial model and e.g. stochastic inputs.
> - **(About Prediction model)**
> The key novelty of our algorithm is to carefully design the decisions given an exact prediction of the cutoff value to give a better than 2 competitive ratio, which matches our lower bound results. Further, we show that this idea can be extended when the prediction of cutoff value is not exact (e.g., interval prediction or probabilistic prediction). This approximate setting is much more interesting in real world settings and requires a significantly more careful analysis. Prior work has considered much stronger prediction models – e.g. frequency predictions for all values [1]. Note that our prediction model is much weaker – as an approximation to the cutoff threshold can be derived from a set of frequency predictions but not vice-versa.
>
> - **(Related work of online knapsack with predictions)**
> two points: (i) two existing works [1] [2] are for integral knapsack with small weights; their model cannot solve our problems since in OFKP, the weight of each item can be close to knapsack capacity; (ii) our prediction model is simple, asking for just a single value; the prediction model used in the other works is to predict the histogram of the item values, which is much more complicated.[1] provides bounds on the total weights for every unit value in advance in predictions.  Such predictions are very stronger than our setting, since they give advanced knowledge of the total weight of items with unit values above the critical value $\hat{v}$, allowing us to adjust the amount we wish to accept at the critical value..
>
>
>
>
> [1] Im et al., *Online knapsack with frequency predictions.* NeurIPS 2021.
>
> [2] Balseiro et al., *Single-leg revenue management with advice.* EC 2023.
>
> [3] John Noga, et al.. *An online partially fractional knapsack problem.* 8th International Symposium on Parallel Architectures, Algorithms and Networks (ISPAN'05), 2005.
>
> [4] Hyun-Woo Kim, et al. *Dynamic bandwidth provisioning using ARIMA-based traffic forecasting for Mobile WiMAX.* Computer Communications 34.1, 2011.
>
> [5] B. Sun, A. Zeynali, T. Li, M. Hajiesmaili, A. Wierman, and D.H.K. Tsang.
> *Competitive algorithms for the online multiple knapsack problem with application to electric
> vehicle charging.* Proc. ACM Meas. Anal. Comput. Syst., 2021.

---

> > ### Comment · Reviewer_G3xe · 2023-11-22
> >
> > The motivation of online fractional knapsack problem looks interesting. I recommend the author to discuss it in detail in the paper. But I am still not convinced by the prediction model. I do not think it is a good idea to directly involve the information of optimal solution in the prediction. I also think the good prediction should not contain complete information of input, such as the frequency predictions. On the other hand, consider the case we have the frequency information as the prediction, where we can compute the cutoff threshold from this prediction. When we consider unreliable predictions, can we derive the “approximated” threshold prediction by the “approximated” frequency prediction? It might be a different thing.

---

### Official Review · Reviewer_BBFM · 2023-10-31

**Soundness:** 3 good
**Presentation:** 3 good
**Contribution:** 2 fair
**Rating:** 6
**Confidence:** 4

**Summary:**

This paper considers the online fractional knapsack problem in the context of three novel prediction models.  For this problem, the optimal solution is characterized by two parameters $\hat v$ and $\hat \omega$, which are the smallest unit value in the optimal solution and the total weight of items with unit-value $\hat v$ in the optimal solution, respectively.  Given this information up front, the online algorithm can simulate OPT exactly - just accept all items with value $> \hat v$ fully and only the first $\hat \omega$ weight of items with value $\hat v$.  The three models considered in this paper assume no knowledge of $\hat \omega$ and progressively noisier information about $\hat v$.

The first model assumes perfect knowledge of $\hat v$.  In this case, the challenge is that we don't know $\hat \omega$ so we don't know how much to allocate to items with value $\hat v$.  A lower bound of $1+ \hat \omega$ is shown for this version of the problem for any (possible randomized) online algorithm.  In the worst case we have $\hat \omega \leq 1$, so this motivates the following strategy for a 2-competitive algorithm.  For each item with value $> \hat v$, allocate half of it.  For items with value $= \hat v$, allocate at most half of these items, up to a total weight of 1/2 the knapsack.  In the case that there is a unique item with value $\hat v$ and weight $\hat \omega$ an improved algorithm with competitive ratio $1+ \hat \omega$ can be given.

The second model assumes that the algorithm is given $\ell \leq u$ s.t. $\ell \leq \hat v \leq u$.  In this case, they treat the items with value in the range $[\ell , u]$ as a separate instance and give these algorithms to a worst-case robust algorithm (the best such algorithms due to prior work have competitive ratio $\alpha = 1+ \ln(u/\ell)$.  The allocation is then split between the two "sub-instances" ($\alpha/(\alpha + 1)$ given to items with value in $[\ell, u]$ and $1/(\alpha + 1)$ given to items with value $> u$.  The overall competitive ratio becomes $1 + \alpha = 2 + \ln(u/\ell)$.

The third model is similar to the second, but the guarantee only holds with probability at least $1-\delta$.  If the guarantee doesn't hold, then the predictions can be arbitrarily wrong (i.e., $\hat v$ could be $\ll \ell$ or $\gg u$).  Thus, the authors propose combining the prior result with the robust algorithm applied to the entire instance (which has values in the range $[L, U]$).  This gives a robust and consistent algorithm for this case.

Finally experiments on simulated instances with values and weights each drawn from a  (bounded above and below) heavy tailed distribution.

**Strengths:**

- The problem and prediction models are well motivated.
- The paper is clearly written and easy to follow.
- The experimental results are promising.

**Weaknesses:**

- Many of the algorithmic ideas are standard in the learning-augmented algorithms literature, e.g., reserving some of the allocation for a robust algorithm as used in Algorithms 4 and 5.
- The comparison with prior work could be clarified further, see below.
- Some details for the experiments could be clarified.  Real data and more realistic predictions could be considered in the experiments.  See below for more discussion.

**Questions:**

## Major Comments and Questions
- Can you provide a more clear comparison with Im et al. 2021?  This paper also extends their results to generalized one-way-trading and it seems possible to extend their model to the fractional setting.  Additionally, it might be nice to compare their algorithm experimentally.
- The probabilistic predictions model is interesting.  Do any lower bound results hold for this model?  In particular is this the optimal tradeoff between robustness and consistency?  Finding optimal trade-offs between robustness and consistency has been of interest lately, see e.g. [1, 2] below.
- Can you clarify exactly how the predictions were created for the experiments for prediction models 2 and 3?  There are many ways to satisfy the promise each of these models guarantee but it seems to me that some ways could be more favorable for your algorithms than others.  For example do we observe different behavior for IPA if $\ell = \hat v < u$ or if $\ell < \hat v = u$?
 - In Figure 2 (a) and (b), the figure seems to cutoff the maximum lines for some of the box plots, can you provide the empirical worst-case results for these experiments as a table or clarify this in the caption?
 - It would be nice to see experiments on real data or with less synthetic predictions derived from the data.

## Minor Comments
 - In Algorithm 3, Line 3 the markup for $b$, $s$ looks different from the rest of the algorithm


### References

[1] - Alexander Wei and Fred Zhang. Optimal robustness-consistency trade-offs for learning-augmented online
algorithms. Advances in Neural Information Processing Systems, 33:8042–8053, 2020.

[2] - Jin, B. and Ma, W. Online bipartite matching with advice: Tight robustness-consistency tradeoffs for the two-stage model. In Oh, A. H., Agarwal, A., Belgrave, D., and Cho, K. (eds.), Advances in Neural Information Processing Systems, 2022.

---

> ### Author Response · Authors · 2023-11-16
>
> Thank you for your review and suggestions, we will incorporate them into our revisions.  Please find our response below:
>
> - **(algorithmic novelty)**
> The key novelty of our algorithm is to carefully design the decisions given an exact prediction of $\hat{v}$ to achieve a competitive ratio better than 2 which matches the lower bound results. We further show that this idea can be extended when the prediction is not exact (e.g., interval prediction or probabilistic prediction), which is more natural in real-world applications. Although the extensions are based on existing algorithmic frameworks in the literature, we show that our new approach can be indeed integrated into the existing framework while attaining good performance guarantees. Moreover, our results are not required to have a bounded value-weight ratio assumption, unlike almost all prior work on these problems.
> - **(Comparison with [1])**
> Thank you for the valuable suggestion. We believe that the frequency prediction model studied in [1] is stronger than ours, in the sense that it provides bounds on the total weights for every unit value in advance.  Such predictions could yield a better competitive ratio in our setting, since they give advanced knowledge of the total weight of items with unit values above the critical value $\hat{v}$, allowing us to adjust the amount we wish to accept at the critical value. By employing their prediction system, we can compute $\hat{v}$, the largest value for which the cumulative weight, including itself and all greater values, surpasses the knapsack size.
> In our setting, we opted to constrain our approach and consider a comparatively weaker prediction model which directly predicts $\hat{v}$.  It is possible that such simpler prediction models would be easier to learn and deploy in practice.
> - **(probabilistic predictions model)**
> Lemma 4.3 establishes the relationship between consistency and robustness of our proposed PIPA algorithm (Algorithm 5).  While we have not explicitly explored lower bounds for this prediction model, we can claim that PIPA is on the Pareto frontier at two points. When γ = 1 and ƍ = 0, we match the optimal consistency bound when predictions are always correct, since we are exactly c-competitive, where c is the optimal competitive ratio of the inner algorithm with perfect predictions.  On the other hand, for any value of ƍ, when γ = 0, we recover the optimal robustness bound, which is exactly the $\ln(U/L) + 1$ bound shown by [2].
> It would be very interesting to explore whether our algorithm achieves the optimal consistency-robustness tradeoff within this probabilistic interval predictions framework for arbitrary values of ƍ and γ.  Thank you for the suggestion!
> - **(experimental setup)**
> We will further clarify our experiment setup in the appropriate sections. For Prediction Model 2, we initially generate an instance of the problem and solve it offline. We obtain the critical value $\hat{v}$ from the offline solution. We then uniformly randomly generate an interval containing the critical value, with size averaging x% of the entire unit value range. For instance, if x = 10, the size of the generated interval will be 10% of the unit value range $[L, U]$.
> For Prediction Model 3, we created an instance as previously and use a coin toss with success probability $p$ to determine whether to provide a generated interval as previously, or a uniformly random interval that does not contain the critical value.
> In the general case, the specific location of the critical value within the interval does not seem to have a large impact, as the inner algorithm for the interval can solve the problem independently. However, there is significance in how many high-valued items fall outside the interval. Algorithm 4 is designed to mitigate this impact by attempting to ensure that a consistent fractional amount is taken from every item (1/(competitive ratio)), thereby minimizing potential disruptions from an incorrect prediction which underestimates the unit values in the sequence.
> - **(experiments on real data)**
> We appreciate the suggestion regarding experiments on real data or with less synthetic predictions derived from the data. While we acknowledge the importance of real-world experiments, most related work in this field has primarily presented numerical results without explicit datasets. Additionally, the utilization of real data often introduces its own set of challenges, including subjectivity in building predictions. We recognize the value of incorporating real-world data in future investigations, and we will look to do so in subsequent iterations of our work.
>
> [1] S. Im et al. Online knapsack with frequency predictions. In Advances in Neural Information Processing Systems (NeurIPS), 2021.
>
> [2] B. Sun et al. Competitive algorithms for the online multiple knapsack problem with application to electric vehicle charging. Proc. ACM Meas. Anal. Comput. Syst., 2021.

---

> > ### Comment · Reviewer_BBFM · 2023-11-22
> >
> > Thank you for the clarifications, especially regarding. the comparisons with [1] and and the experimental setup.  Overall, my evaluation remains the same.

---

### Official Review · Reviewer_SiKU · 2023-11-02

**Soundness:** 3 good
**Presentation:** 2 fair
**Contribution:** 3 good
**Rating:** 6
**Confidence:** 4

**Summary:**

This paper discusses the learning-augmented version of online fractional knapsack problem under different prediction models. In the classical version of the problem, the online algorithm is given the unit value $v_i$ and maximum weight $w_i$ of the $i$-th item at each step. It’s goal is to select the appropriate quantity $x_i\leq w_i$ from each item, such that the total value is maximized without the total weight exceeding the capacity of the knapsack (assumed to be 1 wlog). The authors propose algorithms that utilize predictions on the least valuable item chosen by the optimal (and greedy) offline algorithm to achieve a constant competitive ratio, which ,according to lower bounds in prior work, is impossible without these predictions or other assumptions.

Specifically, in the first prediction model, the algorithm is given the minimum unit value $\hat{v}$ among all items chosen by the offline algorithm, but does not have any knowledge of the quantity that the optimal offline algorithm takes from that item. Because of this, the online algorithm still does not know what decision to make when this particular item appears. The authors address this by having the online algorithm allocate only half of the maximum capacity for each item, while making the decision according to the advice leading to a 2-competitive algorithm. This is later improved to an $(1+\hat{w})$-competitive algorithm, where $\hat{w}$ is the available quantity of the item with unit value $\hat{v}$. The way this is done is by utilizing the fact that the quantity $\hat{w}$ becomes known after the item with unit value $\hat{v}$ arrives.

Subsequently, a different prediction model is considered, in which the prediction is given as an interval $[\ell,u]$ where the item with the lowest unit value that is chosen by the optimal offline algorithm lies. Using this model, and assuming black box access to an $\alpha$-competitive algorithm that works on the promise that all unit values are within a given interval, the authors get a $2+ln(u/\ell)$-competitive algorithm. This is also generalized by adding an error probability for the prediction interval. Finally, the results are supported by experiments comparing the different algorithms and prior work.

**Strengths:**

Overall, I believe the paper is an interesting addition to the learning-augmented algorithms and does address a problem that had not been studied in this context before. The possibility of unreliable predictions is also addressed using interval predictions and introducing prediction error probability.

**Weaknesses:**

There is some sloppiness in the presentation, which can be improved (see below):

-In the end of section 2.1, it’s not clear to me why a linear program formulation for the offline fractional knapsack problem is given, since that problem can be solved by a greedy algorithm and the linear program does not seem to be used elsewhere in this paper. Perhaps it should be emphasized that it is provided only for the purpose of defining the problem.
- Algorithm 2 seems to be suboptimal, since replacing line 7 with “$x_i=w_i$” can only increase the value of the solution. I do realize that this would not affect the worst case competitive ratio of 2 though.


Minor comments:
-In page 4, line 24: part of the sentence is missing
- In Algorithm 2, there is no initialization for the variable z (i.e setting $z=0$ before the while loop)
- In Algorithm 3, line 12: I haven’t checked thoroughly, but something looks wrong here. The reason is that to the left of the minus sign, the expression represents per unit of weight, while it should be representing just cost like the expression on the right of the minus sign.

**Questions:**

Can you comment on the performance on the algorithm when the single value prediction in the first model is not completely accurate?

Is there a smoothness type result where the competitive ratio that depends on additive prediction error (rather than multiplicative, which is in some sense addressed with the second prediction model)?

---

> ### Author Response · Authors · 2023-11-16
>
> Thank you for your detailed and positive review, and your suggestions. We will take them into account while revising the paper.  Please find our response below:
>
> - **(LP definition)**
> You are correct that in Section 2.1, we give an LP formulation only to illustrate the (offline) problem in a concise manner and convey to the reader the meaning of the problem parameters (e.g. decision variables $x_t$). It is not meant to indicate usage of LP solvers. We will clarify this in the writing.
>
> - **(Algorithm 2 (PPA-n))**
> Thanks -- you are completely right that replacing line 7 with $x_i = w_i$ can help. However, if we do this, then we also need to take into account the 'extra capacity' spent on high-value items when determining how much of $\hat{v}$ to include in line 9. If we do not do this, then we may achieve competitive ratio worse than two. For example, consider the following sequence of items, where each item is represented by a (value, weight) pair: (10, 0.5), (1, 1), (10000, 0.49). The optimal offline solution will accept 0.5 of the first item, 0.01 of the second item, and 0.49 of the final item. Note that $\hat{v} = 1$.
> If line 7 is replaced with $x_i = w_i$, PPA-n will accept all of the first item and half of the second item, leaving no remaining capacity to accept the third (most valuable) item. This results in a competitive ratio which can be arbitrarily worse than 2.
> This trade-off between accepting too much early on and leaving enough space to accommodate high-value items which arrive later is the key insight into our design and analysis of PPA-a (Algorithm 3).
> Observe that in PPA-a, in line 8 we indeed set $x_i = w_i$ and in line 12, when admitting the item $\hat{v}$, we take into account the total capacity s of admitted items with higher values.
> Algorithm 2 is meant to serve as a simple warm up to this more optimal, but somewhat more complex procedure.
>
> - **(Minor comments)**
> We will revise Algorithm 2 to initialize the variable $z = 0$ in the first line.
> Regarding line 12 in Algorithm 3 we believe our pseudocode is formally correct, we will clarify the presentation of the PPA-a algorithm.  The primary intuition for line 12 originates in the proof of Theorem 3.4, which was deferred to the Appendix for space purposes.  The high-level idea is as follows: we aim to set $x_i$ such that we accept exactly enough of the critical valued item to guarantee a competitive ratio of $(1 + \hat{\omega})$ in the event that the current item is the final item.  Informally, this will be the amount that we expected to take if the critical valued item arrives first, minus any extra selections made in previous time steps (i.e. items of strictly better unit value which have already been accepted).
>
> - **(Q1: the performance when the single value prediction is not completely accurate)**
> Since the setting we study is adversarial, there will be no competitive ratio guarantee for Algorithms 2 and 3 if predictions are not accurate. This motivates our development of IPA and PIPA (Algorithms 4 and 5), which are able to “hedge” against untrusted / inaccurate predictions, and use a similar approach as PPA-a (Algorithm 2) to solve IPA(Algorithm 4) which allow us to have prediction over range $\[\ell,u\]$, moreover, we can use algorithm 2,3,4 in PIPA as subroutine. Algorithms 2 and 3 can be thought of as intermediate results building up to these more robust algorithms.
>
> - **(Q2)**
> We can model arbitrary error in the interval prediction model, whether the prediction error is additive or multiplicative. For additive error, we can find the interval $[\hat{v}-e, \hat{v}+e]$, and for multiplicative error, we can find the interval $[\hat{v} \times (1+e), \hat{v} \times \frac{1}{1+e}]$. The only difference between these two cases would be the rate of increase in the competitive ratio, which is $\ln(u/\ell) + 2$.

---

### Official Review · Reviewer_Qu7e · 2023-11-02

**Soundness:** 2 fair
**Presentation:** 3 good
**Contribution:** 2 fair
**Rating:** 3
**Confidence:** 2

**Summary:**

The paper studies the online fractional knapsack problem in which items arrive online. Upon the arrival of a new item, we need to decide what fraction of the item to accept. In turn, we incur some weight and gain some value both proportional to the fraction of the item we accepted. Items come as a pair of values for their unit weight and value numbers.

Although the knapsack problem is to some extent motivated by the applications mentioned in the paper, the assumption that a fraction of an item can be accepted makes the problem less interesting and not very applicable to these settings. When serving online traffic (queries or tasks), it is not acceptable to serve a subset of tasks. Load assignment is different from knapsack optimization. In online advertising, it is not reasonable to ask the sales team of an advertisement company to spend a fraction of their budget on your framework.

Also in online advertising, many times we are not facing a truly online problem. Negotiations may occur when we have access to all or a big subset (batch) of the requests/items.

Most of the results are focused on providing competitive ratio algorithms in settings that relatively strong prediction signals are provided. For example when the exact value of the cutoff threshold for the offline optimum algorithm is known. There is some novelty in the analysis when they want to figure out how to allocate their budget to items above this threshold and the ones at this threshold to get a better than 2 approximation. But the rest of the analysis is relatively straightforward given they need to plug in the previous algorithms in the literature and see what happens.

The comparison benchmarks are optimum offline which is standard for online algorithms. However given the strong assumptions such as knowing exactly the cutoff threshold (Theorem 3.1), it may make sense to study the optimum online algorithm and use it for comparison.

The experiments are based on synthesized data.

**Strengths:**

The paper introduces some framework to add predictions to the online fractional knapsack problem.

**Weaknesses:**

Although the knapsack problem is to some extent motivated by the applications mentioned in the paper, the assumption that a fraction of an item can be accepted makes the problem less interesting and not very applicable to these settings. When serving online traffic (queries or tasks), it is not acceptable to serve a subset of tasks. Load assignment is different from knapsack optimization. In online advertising, it is not reasonable to ask the sales team of an advertisement company to spend a fraction of their budget on your framework.

Also in online advertising, many times we are not facing a truly online problem. Negotiations may occur when we have access to all or a big subset (batch) of the requests/items.

Most of the results are focused on providing competitive ratio algorithms in settings that relatively strong prediction signals are provided. For example when the exact value of the cutoff threshold for the offline optimum algorithm is known. There is some novelty in the analysis when they want to figure out how to allocate their budget to items above this threshold and the ones at this threshold to get a better than 2 approximation. But the rest of the analysis is relatively straightforward given they need to plug in the previous algorithms in the literature and see what happens.

The comparison benchmarks are optimum offline which is standard for online algorithms. However given the strong assumptions such as knowing exactly the cutoff threshold (Theorem 3.1), it may make sense to study the optimum online algorithm and use it for comparison.

The experiments are based on synthesized data.

**Questions:**

Read the weaknesses part for the main questions. The rest are here:

Other comments:
Page 1: “It is well known that no deterministic …”, what about randomized algorithms?

Section 2.2, “The threshold-based algorithm is shown in Algorithm 1.” I suggest adding a citation to Sun et al. in this sentence to make it clear that Alg 1 is their algorithm and not yours.

Paragraph after Prediction Model II: “The quality of prediction in this c u - ell …”. There seems to be a typo here.

Algorithm 3 pseudocode: in line 10, add a comment that at this point \hat{w} is revealed since it is the weight of this new item. BTW, do you have any justification that the values of items are distinct? Line 12 seems to have some extra parenthesis, you probably get a compilation error with this.

---

> ### Author Response · Authors · 2023-11-16
>
> Thank you for the detailed review and for your suggestions. We will take them into account while revising the paper.  Please find our response below:
>
> - **(Motivation for studying online fractional knapsack problem)**
> While it is true that online fractional knapsack (OFKP) doesn't address all online knapsack problems, there are numerous problems that can be effectively modeled by OFKP. This includes, for example, dynamic bandwidth allocation in wireless or communication networks [1, 2], online electric vehicle charging in smart grids [4], and online trading problems in financial markets [4]. For bandwidth allocation problems, multiple users share the same bandwidth with fixed capacity. Upon the arrival of each user, it requests for at most $w_i$ amount of bandwidth and each unit of bandwidth has a value $v_i$. The system operator determines how much bandwidth is allocated to the user with the objective of maximizing the total profit of the allocated bandwidth. OFKP exactly models this problem and the integral version of the online knapsack cannot provide a meaningful solution, especially when $w_i$ is comparable to the bandwidth capacity.
> We do agree that it would be interesting to extend our approach to integral knapsack problems – often results for the fractional case can be extended to the integral case if we assume that the item sizes are small enough. However, detailed derivation of the results needs further analysis which is beyond the scope of current work.
>
> - **(About the algorithmic novelty)**
> The key novelty of our algorithm is to carefully design the decisions given an exact prediction of the cutoff value to give a better than 2 competitive ratio, which matches our lower bound results. Further, we show that this idea can be extended when the prediction of cutoff value is not exact (e.g., interval prediction or probabilistic prediction). This approximate setting is much more interesting in real world settings and requires a significantly more careful analysis. Prior work has considered much stronger prediction models – e.g. frequency predictions for all values [5]. Note that our prediction model is much weaker – as an approximation to the cutoff threshold can be derived from a set of frequency predictions but not vice-versa.
>
> - **(About the comparison benchmark)**
> Assuming by optimal online you mean matching upper bound and lower bound for the algorithm. We would like to clarify that without prediction Algorithm 1 is optimal which we used in comparisons and given the prediction of the cutoff value, we have already derived the optimal online algorithm(Algorithm 2,3) that can achieve the best possible competitive ratio.
>
> - **(Additional clarifications)**
> Thanks to the reviewer for their suggestions on the presentation. We clarify those points below and will revise the paper accordingly. (i) Randomized algorithms also cannot achieve a bounded competitive ratio for the most general online (integral) knapsack problem without additional assumptions, which has been shown in [3]. (ii) We will add a citation in Section 2.2 to further clarify that the threshold-based algorithm (Algorithm TA) is due to [4]. In fact, we have cited this paper when presenting Lemma 2.1 that claims the performance of Algorithms. (iii) we will address the typos in Prediction Model II and adopt your suggestion to include an explanatory comment on line 10 of Algorithm 3.
>
> - **(Justification on algorithm 3 solves unique values problem)**
> If one perturbs the input values by an arbitrarily small amount, then uniqueness can be ensured while keeping the objective function the same up to an arbitrarily small error. Thus, our results for interval predictions in $\[\ell,u\]$ (with $u/\ell$ arbitrarily close to 1) indeed capture the non-unique value setting. We will add an explanation of this in the paper. We do think that extending our $1+\hat{w}$ competitive ratio result to the non-unique value/interval settings would be interesting. In fact, we are actively working in this direction, and have a conjectured algorithm but an incomplete analysis.
>
> [1] John Noga, et al.. *An online partially fractional knapsack problem.* 8th International Symposium on Parallel Architectures, Algorithms and Networks (ISPAN'05), 2005.
>
> [2] Hyun-Woo Kim, et al. *Dynamic bandwidth provisioning using ARIMA-based traffic forecasting for Mobile WiMAX.* Computer Communications 34.1,2011.
>
> [3] Y. Zhou, D. Chakrabarty, and R. Lukose. *Budget constrained bidding in keyword
> auctions and online knapsack problems.* In Proceedings of the 17th International Conference
> on World Wide Web, pp. 1243–1244,2008.
>
> [4] B. Sun, A. Zeynali, T. Li, M. Hajiesmaili, A. Wierman, and D.H.K. Tsang.
> *Competitive algorithms for the online multiple knapsack problem with application to electric
> vehicle charging*, 2021.
>
> [5] S. Im, R. Kumar, M. Montazer Qaem, and M. Purohit, *Online Knapsack with Frequency Predictions,* in Advances in Neural Information Processing Systems, 2021.

---

> > ### Comment · Reviewer_Qu7e · 2023-11-21
> > **Motivation of the problem by bandwidth allocation**
> >
> > Thanks for providing more context on the motivation. The only one you elaborate on is bandwidth allocation. I see you cite [1] and [2] papers. The paper [1] seems to be purely theoretical without any connection to a real instance. Paper [2] has some more details but at its core it seems to be a traffic aware allocation problem. I don't think you are allowed to come up with any fractional solution that might dismiss some components of the allocation problem. Since the motivation still seems problematic, I suggest doing a more comprehensive review to provide a solid application. I have seen many papers devoting one or two pages just to provide the exact formulation of the applications they are focusing on. To summarize, I am still not convinced this is an important enough problem for ICLR community.

---

### Meta-Review · Area_Chair_asvg · 2023-12-07

**Metareview:**

The paper studies the online fractional knapsack problem with predictions. This work introduces and studies 3 prediction models for the minimum value of items included in the optimal solution, and designs algorithms that incorporate the predictions.

This paper had some support and some of the reviewers appreciated some of the aspects of the models introduced in this work. There remained concerns regarding the motivation for the prediction models considered and the potential applications of this works. Additionally, the technical novelty of this work is very limited. Overall, this paper does not meet the threshold for acceptance.

**Justification For Why Not Higher Score:**

The conceptual and technical novelty is very limited. The problem studied is relevant, but the motivation for the prediction models considered is unclear.

**Justification For Why Not Lower Score:**

N/A

---

### Decision · Program_Chairs · 2024-01-16

Reject